# Chemical tools for epichaperome-mediated interactome dysfunctions of the central nervous system

Alexander Bolaender[1,11], Danuta Zatorska[1,11], Huazhong He[1,11], Suhasini Joshi[1,11], Sahil Sharma[1,11], Chander S. Digwal[1,11], Hardik J. Patel[1], Weilin Sun[1], Brandon S. Imber [1], Stefan O. Ochiana[1], Maulik R. Patel[1], Liza Shrestha[1], Smit. K. Shah[1], Shuo Wang[1], Rashad Karimov [1], Hui Tao[1], Pallav D. Patel[1], Ananda Rodilla Martin[1], Pengrong Yan [1], Palak Panchal[1], Justina Almodovar[1], Adriana Corben[2], Andreas Rimner [3], Stephen D. Ginsberg [4,5], Serge Lyashchenko[6], Eva Burnazi[6], Anson Ku[7], Teja Kalidindi[7], Sang Gyu Lee[7], Milan Grkovski [8], Bradley J. Beattie[8], Pat Zanzonico[8], Jason S. Lewis [6,7], Steve Larson[7], Anna Rodina[1,12], Nagavarakishore Pillarsetty [7,12], Viviane Tabar[9], Mark P. Dunphy [7], Tony Taldone[1,12], Fumiko Shimizu [1,9✉] & Gabriela Chiosis [1,10,12✉]

Diseases are a manifestation of how thousands of proteins interact. In several diseases, such as cancer and Alzheimer's disease, proteome-wide disturbances in protein-protein interactions are caused by alterations to chaperome scaffolds termed epichaperomes. Epichaperome-directed chemical probes may be useful for detecting and reversing defective chaperomes. Here we provide structural, biochemical, and functional insights into the discovery of epichaperome probes, with a focus on their use in central nervous system diseases. We demonstrate on-target activity and kinetic selectivity of a radiolabeled epichaperome probe in both cells and mice, together with a proof-of-principle in human patients in an exploratory single group assignment diagnostic study (ClinicalTrials.gov Identifier: NCT03371420). The clinical study is designed to determine the pharmacokinetic parameters and the incidence of adverse events in patients receiving a single microdose of the radiolabeled probe administered by intravenous injection. In sum, we introduce a discovery platform for brain-directed chemical probes that specifically modulate epichaperomes and provide proof-of-principle applications in their use in the detection, quantification, and modulation of the target in complex biological systems.

[1] Program in Chemical Biology, Sloan Kettering Institute, New York, NY, USA. [2] Department of Pathology, Memorial Sloan Kettering Cancer Center, New York, NY, USA. [3] Department of Radiation Oncology, Memorial Sloan Kettering Cancer Center, New York, NY, USA. [4] Center for Dementia Research, Nathan Kline Institute, Orangeburg, NY, USA. [5] Departments of Psychiatry, Neuroscience & Physiology and the NYU Neuroscience Institute, NYU Grossman School of Medicine, New York, NY, USA. [6] Radiochemistry and Molecular Imaging Probes Core, Sloan Kettering Institute, New York, NY, USA. [7] Department of Radiology, Memorial Sloan Kettering Cancer Center, New York, NY, USA. [8] Department of Medical Physics, Memorial Sloan Kettering Cancer Center, New York, NY, USA. [9] Department of Neurosurgery, Memorial Sloan Kettering Cancer Center, New York, NY, USA. [10] Breast Cancer Medicine Service, Memorial Sloan Kettering Cancer Center, New York, NY, USA. [11]These authors contributed equally: Alexander Bolaender, Danuta Zatorska, Huazhong He, Suhasini Joshi, Sahil Sharma, Chander S. Digwal. [12]These authors jointly supervised this work: Anna Rodina, Nagavarakishore Pillarsetty, Tony Taldone, Gabriela Chiosis. ✉email: fshimizu@gmail.com; chiosisg@mskcc.org

Most proteins do not act alone – they interact with others in the cellular milieu and it is the architecture of these combined interactions that defines the activity of protein pathways, and in turn the cellular phenotype[1]. Detecting and modulating aberrant protein–protein interactions (PPIs) in the context of disease is therefore of critical importance as it may lead to biomarkers and drugs that detect and target dysfunctions in proteome-wide connectivity (i.e., edgetic perturbations within interactome networks) rather than individual gene or protein defects[2–5].

The chaperome, a large assembly of >300 chaperones, co-chaperones, and related factors[6], is considered a safeguard of proteome function and a regulator of protein assembly[7]. Through low affinity and transient interactions, chaperome members interact with components of the proteome with the end goal of folding proteins, transporting them to a proper cellular location or facilitating their assembly into protein complexes[8–11]. The chaperome is one of the most abundant protein assemblies in human cells and in turn in the human body, also prevalent in all cell types and tissues[12]. Cellular stressors appear to disturb the dynamic nature of such interactions, and stabilized oligomeric chaperone species, later dubbed as epichaperomes, were reported on Native PAGE in cells exposed to stressors such as heat stress, glucose deprivation, or a toxin such as antimycin A[7]. Such stable and soluble heterooligomeric chaperome pools, also identified in tumors, Parkinson's disease (PD) neurons, and Alzheimer's disease (AD) brain tissues, work as scaffolding platforms rather than in folding[7]. Heat shock protein 90 (HSP90) and heat shock cognate 70 (HSC70) chaperones play a central role in the formation of these epichaperome structures, yet it should be noted there is a fundamental – structural, dynamic and functional – difference between the epichaperome and chaperones[7,13–17]. Epichaperomes are scaffolds – they rewire the connectivity and function of protein networks by remodeling how thousands of proteins interact in conditions of chronic cellular stress (i.e. for example in diseases such as cancers, AD, and PD)[7,14,16]. Conversely, chaperones, co-chaperones, and their complexes interact with a protein to process it through the chaperone folding cycle[18,19]. Epichaperomes are long-lived oligomers of chaperome members[7,13–17]. Conversely, chaperones interact dynamically with one another and with client proteins on the millisecond to second timescale to make folding versus degradation decisions through transient PPIs within the context of the proteostasis network that are central to maintaining the cellular proteome. Epichaperomes are specific to cells exposed to specific stressors[7,13–17]. Conversely, chaperones are highly abundant and ubiquitous proteins[12]. The fraction of HSP90 and HSC70 incorporated into epichaperomes is minor when compared to the pool involved in folding functions[7,13–17]. In addition, and unlike the ubiquitous chaperones, epichaperomes are localized to diseased cells and tissues[20,21].

A causal link between epichaperome formation and proteome-wide connectivity dysfunction was demonstrated in cancer, PD, and AD[13,14,16,17]. Through their scaffolding function, epichaperomes pathologically rewire PPIs in these diverse diseases at the proteome-wide level. They cause thousands of proteins to improperly organize inside cells, aberrantly affecting cellular phenotypes[7]. This realization posits epichaperomes as pathologic regulators of cellular stress of high significance in disease biology, and as surrogates for aberrant molecular interactions in disease. Therefore, the availability of chemical probes to study the function of these assemblies as well as drug candidates to dismantle them are of high significance.

Protein structure, dynamics, and function are interdependent, and in this sense, the chaperones HSP90 and HSC70 become distinct entities when part of the epichaperome. In their folding function, chaperones form short-lived dynamic complexes with individual proteins. Conversely, epichaperomes are stabilized complexes comprising multiple chaperones and co-chaperones. These features provide an opportunity for the specific interaction of small molecules through kinetic selectivity[22,23] to discriminate the epichaperomes from the more abundant chaperome proteins[15,21]. One such small molecule is PU-H71 (1). While initially discovered as an HSP90 inhibitor[24], later studies suggested it kinetically prefers a tumor enriched pool[25–28] that was subsequently identified to be the epichaperome[21]. It dissociates from epichaperomes much slower ($k_{off}$ of several hours) than from other cellular pools of HSP90 ($k_{off}$ of minutes)[13,15]. PU-H71 and associated probes have therefore played an important role in deciphering the causative link between disease-inducing stressors, epichaperome formation, and proteome-wide interactome network dysfunction[14,17,21]. PU-H71 however does not permeate the blood–brain barrier (BBB) rendering it of limited use in the study and treatment of diseases of the central nervous system (CNS).

The BBB plays an important role in maintaining homeostasis in the CNS by restricting the transport of potentially toxic molecules. The ability to deliver agents into the CNS is therefore severely hindered by the BBB, often resulting in minimal bioavailability of drugs and presents as a major obstacle to detecting and treating CNS disorders[29]. The BBB is composed of a single layer of endothelial cells connected by tight junctions. Brain microvascular endothelial cells lack fenestrations, have few pinocytotic vesicles, and express a variety of metabolic enzymes and membrane efflux transporters, such as P-glycoprotein (Pgp)[30]. These features make the BBB a formidable barrier that small molecules must overcome to reach the brain parenchyma, and it is estimated that >98% of organic small molecules do not cross the BBB[31].

To overcome such limitations, we here introduce a discovery platform – from structural and biochemical insights to in vivo testing – for the discovery of BBB-permeable epichaperome probes. We provide proof-of-principle both in mice and humans that the platform may yield a toolset consisting of probes to study, detect, and treat epichaperome-mediated interactome dysfunctions of the CNS. We here identify PU-HZ151 and radiolabeled PU-HZ151 for which we provide cellular and organismal level evidence for their selectivity for epichaperomes over the individual chaperome members, and for their ability to productively engage the target in cells, mice, and humans. We also provide proof-of-principle showcasing the use of the toolset in disease study, using brain cancer as a model. In addition to classical models of disease, the toolset is applicable to human patients, defining them as translational agents. Supportive of their significance, these probes are now studied in clinical trials in human patients to detect and treat diseases of the CNS (NCT03371420, NCT03935568, NCT04311515, NCT04505358, and NCT04782609).

## Results

**Epichaperome probe design**. HSP90 is a key component of epichaperomes and PU-H71 interacts with epichaperomes through insertion into the N-terminal domain pocket of the HSP90[13]. This pocket is highly flexible and in its free state exists in different conformations[32]. Crystal structures of HSP90 with a variety of small molecule ligands also show a high plasticity in binding, particularly in helix 3. For example, PU-H71 bound to HSP90α unveils a subpocket in this region, with the 8-aryl moiety of PU-H71 exposing a lipophilic cavity whereby the phenyl ring becomes stacked between Phe138 and Leu107 (PDB:2FWZ)[33]. This so-called 'helix' conformation binding mechanism (Fig. 1a) was observed for some but not all HSP90-ligands[32]. For example, geldanamycin (2) prefers a so-called 'loop' pocket conformation

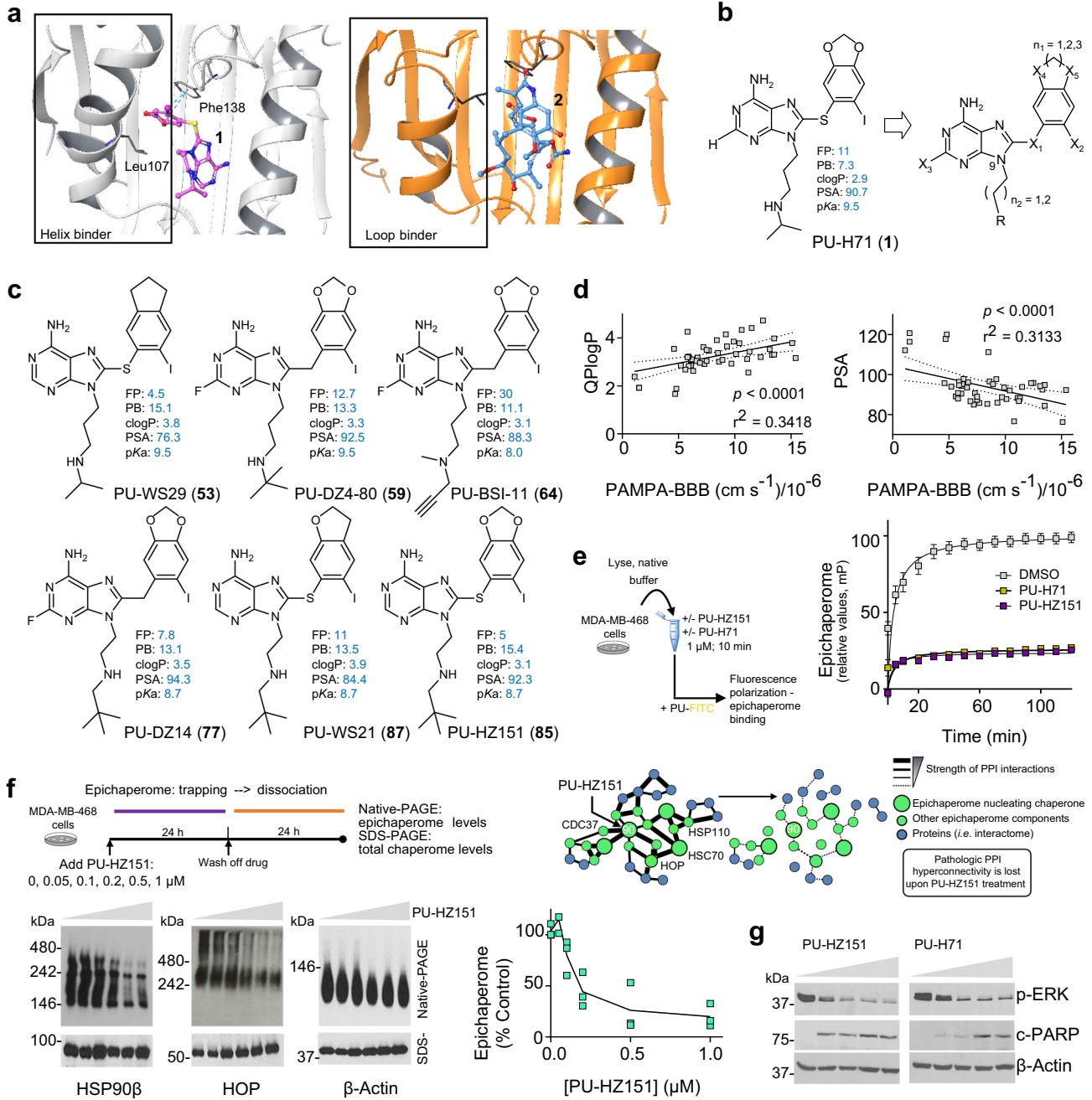

**Fig. 1 Design and discovery of the CNS-directed epichaperome probes. a** HSP90 crystal structure in complex with PU-H71 (1) (PDB ID: 2FWZ) or geldanamycin (2) (PDB ID:1YET) representing helical and loop binding conformations, respectively. **b** Schematic showing proposed sites for modification. **c** Chemical structure of potential CNS-targeted epichaperome probes and their determined affinity and permeability characteristics. Experimentally determined PAMPA-BBB (PB) permeability efficient (Pe), $10^{-6}$ cm s$^{-1}$ and fluorescence polarization (FP) EC50, nM values and calculated PSA, polar surface area, Å$^2$; acid dissociation constants, pKa and the logarithmic value of the 1-octanol/water partition coefficient (log P) are also shown. See also Supplementary Figs. 1–16 and Supplementary Table 6. **d** Correlative analysis between calculated compound properties and experimentally determined BBB permeability. Pearson's r two-tailed, $n = 47$ individual compounds. **e** Competitive binding of PU-HZ151 to epichaperomes in MDA-MB-468 cell homogenates. Graph, mean ± s.e.m. of three replicates. **f** Biochemical analysis of epichaperome modulation by PU-HZ151 in the epichaperome-positive cell line MDA-MB-468. Representative gels and graphed data ($n = 3$ individual data points are shown). Native PAGE shows the fraction of the chaperones incorporated into epichaperomes. SDS-PAGE shows the total chaperone levels. **g** Western blot analysis of MDA-MB-468 cells treated for 24 h with PU-HZ151 or PU-H71 (0, 0.1, 0.25, 0.5, and 1 μM). c-PARP, cleaved PARP. Source data are provided as a Source Data file.

(Fig. 1a). The differential binding mode influences the kinetics of ligand binding – ligands which induce (or capture) the helix conformation, termed 'helix-binders', have lower association and dissociation rate constants for binding to HSP90 than 'loop-binders'[32]. Helix binders can reach two-log slower dissociation

rates when compared to loop binders. The 'helix' conformation, while possibly a stochastic event in free HSP90, it may become enriched in disease when HSP90 becomes incorporated into epichaperomes[7,17,20]. With this in mind, we designed our ligands with PU-H71 as a starting point for medicinal chemistry efforts

aiming to improve brain delivery while retaining favorable affinity. We then profiled lead(s) in cellular and organismal models that endogenously recapitulate the HSP90 conformation and dynamics intrinsic to the epichaperome. This discovery platform is more likely to capture the native configuration of the target, and in turn, result in probes with epichaperome selectivity over abundant HSP90 pools.

**Medicinal chemistry for CNS delivery**. In general, small molecules are likely to pass the BBB by passive diffusion[29,30], and several important physicochemical parameters are associated with such permeability behavior such as $pK_a$ (which determines the charge state of a molecule in a solution of a particular pH), lipophilicity (which determines distribution of a molecule between the aqueous and the lipid environments in the body), solubility (which limits the concentration that a molecule can present to the BBB), and membrane permeability (which determines how quickly molecules can cross membrane barriers separating compartments in the body)[31,34]. For PU-H71 there are three structural elements that could be chemically modified to inquire into the influence of these parameters on BBB-permeability (Fig. 1b). These include modification of (1) the $X_2$-substituent on the aryl ring (Supplementary Table 1), (2) the fused methylenedioxy ring (Supplementary Table 2), and (3) the N9-chain (Supplementary Tables 3 and 4). Specifically, to probe the effect of $X_2$ polarity and size on brain permeability, this was exchanged from iodine to a large spectrum of substituents including bromine, alkyl, alkenyl, aryl, and cyano substituents (Supplementary Table 1). The fused methylenedioxy ring was altered to gain lipophilicity (Supplementary Table 2). The $pK_a$ of the N9-chain amine was modified from secondary to tertiary to quaternary, while its R substituent was altered with chains that span a large spectrum of size and lipophilicity (Supplementary Tables 3 and 4). Derivatives with $X_1$ being either S or $CH_2$ were investigated as these are known to be favorable linkers at such position[24,35,36]. The designed analogs **4** through **87** were synthesized as per the Supplementary Note 1 and Supplementary Figs. 1–11.

**First-pass in vitro filtering**. The synthetic efforts resulted in 48 derivatives, and the initial goal was to rapidly filter out modifications that interfered with a favorable occupancy of the HSP90 pocket. We employed a fluorescence polarization (FP) assay that measures equilibrium competitive binding to the heterogeneous HSP90 pools found in disease-afflicted brains (AD transgenic mice)[37]. For preliminary BBB-permeability assessment we employed a variation of the PAMPA permeability assay designed to predict BBB permeation[38]. It uses a mixture of brain lipids selected to model the composition of the BBB, and is reported to predict BBB penetration by passive diffusion with >90% certainty[38]. In our set-up we used extracted porcine polar brain lipid, and validated the assay using desipramine, as high, and caffeine, as low, PAMPA-BBB permeability controls.

We found replacing iodide on $X_2$ with aromatic rings, but not with aliphatic chains, had a favorable effect on permeability but was unfavorable for affinity (Supplementary Table 1). Exchanging the methylenedioxy with ethyleneoxy, ethylenedioxy, or propyl (eg. **53**, PU-WS29, Fig. 1c and Supplementary Table 2), or modifying the nature of the N9-chain's lipophilicity (eg. **64**, PU-BSI-11 and **59**, PU-DZ4-80, Fig. 1c and Supplementary Table 3), improved permeability while retaining favorable binding. Of note, changing the position of the N9-terminal amine one carbon nearer to the purine ring ($n_2 = 1$ versus 2, Fig. 1b), increased affinity and allowed for the introduction of N-alkyl modifications that engendered favorable permeability (**77**, PU-DZ14; **87**, PU-WS21, and **85**, PU-HZ151, Fig. 1c and Supplementary Tables 4 and 5). We observed a significant positive correlation between

lipophilicity and permeability (clogP versus PAMPA-BBB, $p < 0.0001$, $r^2 = 0.3418$; PSA versus PAMPA-BBB, $p < 0.0001$, $r^2 = 0.3133$, Fig. 1d), albeit derivatives clustered in a narrow range (mean ± SD, PSA = 94.1 ± 8.9; clogP = 3.2 ± 0.6, Supplementary Table 6).

Of these compounds, priority for development was given to derivative **85** (Supplementary Figs. 11 and 12), called PU-HZ151 (EC$_{50}$ of 5 nM in the FP assay as compared to 11 nM for PU-H71 and a logD of 2.37 as compared to 1.21 for PU-H71, Supplementary Table 6). A favorable feature of PU-HZ151 is the endogenous presence of iodine at $X_2$; this naturally occurring stable isotope iodine-127 ($^{127}$I) can be replaced with the positron emitter iodine-124 ($^{124}$I) to provide a chemical probe for use in epichaperome detection by positron emission tomography (PET) imaging or with iodine-131 ($^{131}$I) for detection by autoradiography. These chemical tools could enable detection and quantitation of epichaperomes, and in turn proteome dysfunction, at both cellular and organismal level, in the CNS, in either brain tissue or in live patients. We therefore designed a two-step synthesis for radiolabeling of PU-HZ151 by converting the original molecule into the stannane precursor, followed by iodometallation with [$^{124}$I]-NaI or [$^{131}$I]-NaI (Supplementary Fig. 13 and Supplementary Note 1).

In the PAMPA-BBB we recorded for PU-HZ151 a $P_e$ of 15.4 ± 1.3 × 10$^{-6}$ cm s$^{-1}$, similar to desipramine, the standard of high CNS permeability. For comparison, we measured a $P_e$ of 7.3 ± × 10$^{-6}$ cm s$^{-1}$ for PU-H71 in the same assay. PU-HZ151 demonstrated on-target activity by effectively modulating the epichaperome in both cell homogenates and live cells (Fig. 1e–g). PU-HZ151 was assayed in vitro at 10 μM for off-target binding to 167 substrates (70 in the general side effect profile (SEP) II diversity panel and 97 in the scanEDGE kinase panel) to show little to no off-target binding interactions (Supplementary Table 7 and Supplementary Fig. 14a, b). General SEP II contains brain-resident key proteins, including neurotransmitter receptors, ion channels, ion pumps, synthetic enzymes, and transporter proteins. ScanEDGE focuses on kinases and was used to exclude potential nonspecific interaction of PU-HZ151 (a purine-scaffold agent) with typical ATP-binders, such as kinases. PU-HZ151 showed comparable activity in both Pgp-overexpressing and Pgp-low epichaperome-positive cell lines, suggesting its efficacy is unlikely to be affected by Pgp overexpression in the brain (Supplementary Fig. 15a–c).

The metabolic stability and metabolite characterization of PU-HZ151 were evaluated, and the cytochrome P450 enzymes responsible for the metabolism of PU-HZ151 were identified (Supplementary Tables 8 and 9 and Supplementary Fig. 16a–c). Six putative metabolites were identified in liver microsomes of human, rat, and dog and three metabolites in mouse microsomes. PU-HZ151 is mainly metabolized by 3A, 3A4, and 2C9 and to a lesser extent by 1A2, 2D6, 2B6, 2C8, and 2C19 and is essentially unaffected by 2A6. However, we detected only the intact PU-HZ151 molecule in epichaperome-positive lesions and in the brain after 1 h post-intravenous PU-HZ151 injection, indicating metabolites are both rapidly cleared from the brain and the whole body and/or are epichaperome-inert (Supplementary Fig. 16a–c).

Extended plasma circulation, stability in liver microsomes, and low metabolic clearance are often sought in drug discovery as a mechanism for obtaining good brain exposure. In the discovery and development of epichaperome agents, extended presence of the agent at sites other than those expressing the target is not a goal[13–17] because the slow off-rate of an epichaperome probe from the target assures that, once the ligand binds to the target, the target remains blocked for hours to days[22,23]. Therefore, good $C_{max}$ at the site of target expression and a rapid systemic clearance, are the two parameters we aim for, as recently reported[15].

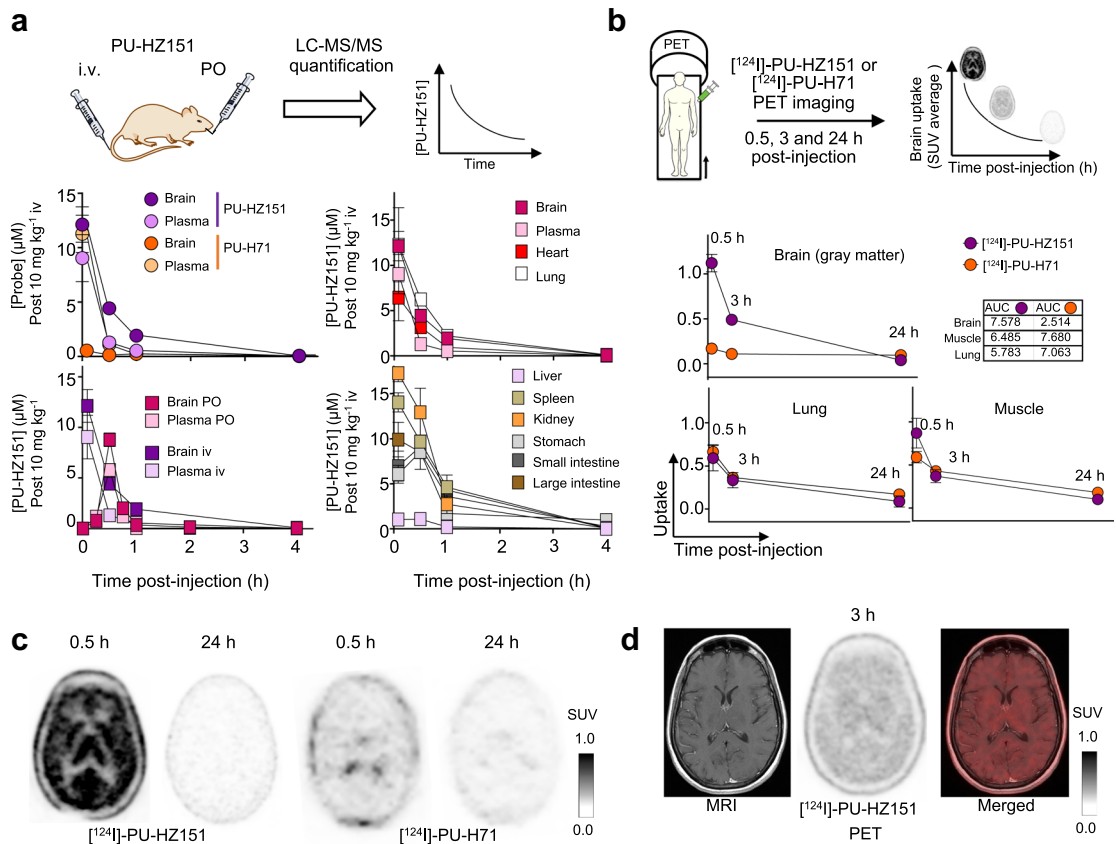

**Fig. 2 Brain permeability of derivative 85, PU-HZ151, in mice and humans. a** Time-dependent molar concentration curve of PU-HZ151 in plasma and brain determined by liquid chromatography-tandem mass spectrometry (LC-MS/MS). Non-CNS, metabolizing and non-metabolizing, organs as well as PU-H71 are shown for comparison. Probes were injected intravenously (i.v.) or by oral gavage (PO) at 10 mg kg$^{-1}$ to healthy male B6D2F1 mice ($n = 12$ per experimental condition, 3 per time point). Data are presented as mean ± s.e.m. See also Supplementary Fig. 17. **b** Experimental design for positron emission tomography (PET) imaging in human patients. Images were obtained at 0.5, 3, and 24 h post [$^{124}$I]-PU-HZ151 or [$^{124}$I]-PU-H71 injection. non-CNS and non-Pgp organs and tissues are shown for reference. Graph, mean ± s.e.m. of $n = 3$ for [$^{124}$I]-PU-HZ151 PET and $n = 14$ at 0.5 h; $n = 24$ at 3 h and 24 h for [$^{124}$I]-PU-H71 PET. AUC area under the curve, SUV standard uptake value. **c** Representative axial brain PET images of normal (non-diseased) human brain obtained as in **b** at 0.5 and 24 h post [$^{124}$I]-PU-HZ151 single injection. PU-HZ151 demonstrates penetration across BBB, with no binding to neurologic structures and complete washout by 24 h. [$^{124}$I]-PU-H71 is shown for comparison. **d** A representative magnetic resonance imaging (MRI) overlaid with the PET image obtained at 3 h post [$^{124}$I]-PU-HZ151 injection shows the anatomic localization of PU-HZ151. Scale bars, PET window display intensity scales, with upper and lower SUV thresholds. Source data are provided as a Source Data file.

**BBB permeability in vivo**. To confirm the superior brain permeability of PU-HZ151 over PU-H71, we next performed studies in mice (Fig. 2a) and in human patients (Fig. 2b–d and Supplementary Note 2). We first monitored brain and plasma exposure to PU-HZ151 and PU-H71 by killing cohorts of animals at selected times post-intravenous (i.v.) injection of 10 mg kg$^{-1}$ of each probe, and then measuring their levels by liquid chromatography-tandem mass spectrometry (LC-MS/MS). We recorded similar plasma exposure for PU-HZ151 and PU-H71 (area under the curve, AUC of 3.52 versus 3.47 μM h$^{-1}$, respectively). Conversely, the brain exposure was approximately ten-fold higher for PU-HZ151 than for PU-H71 (AUC of 8.25 versus 0.89 μM h$^{-1}$, respectively). A favorable brain uptake, equaling that observed for non-CNS and non-Pgp organs, was obtained for PU-HZ151 following both single intravenous and oral administration (Fig. 2a, $F = 68\%$, brain/plasma = 1.6 by oral administration and $C_{max}$ of 10–15 μM observed in both brain and lung). We corroborated these studies with [$^{131}$I]-labeled agents and gamma counting of radioactivity (Supplementary Fig. 17a).

To evaluate the brain permeability of PU-HZ151 into the human brain, we produced the [$^{124}$I]-labeled derivative and performed serial PET imaging studies of human patients ($n = 3$;

NCT03371420), and derived time-activity (i.e. standard uptake values (SUV) versus time post-administration) curves for the brain and various other tissues (Fig. 2b). PET data obtained for [$^{124}$I]-labeled PU-H71 (NCT01269593 and ref. [39]) were used for comparison. These studies provided evidence, both quantitatively (Fig. 2b) and visually (Fig. 2c, d), for a substantially greater uptake of PU-HZ151 over PU-H71 into the brain (SUV of 1.12 versus 0.17 at 0.5 h, respectively). The AUC values recorded for [$^{124}$I]-PU-HZ151 in the human brain post-single injection are at par with what was measured in the human muscle and lung, supportive of uptake of PU-HZ151 into the human brain (Fig. 2b). Magnetic resonance imaging (MRI), by offering structural and soft tissue resolution, confirmed the presence of PU-HZ151 in the brain tissue (Fig. 2d). The primary outcome of the trial was to measure pharmacodynamics, blood samples were also collected 1, 5, 15, 30, 60–90, and 150–270 min post-injection to determine the pharmacokinetic profile of [$^{124}$I]-PU-HZ151 and support a rapid clearance of PU-HZ151 from systemic circulation (Supplementary Table 10), which is optimal for imaging. The secondary outcome of the trial was to measure adverse events following administration of the probe. Safety of [$^{124}$I]-PU-HZ151 in human subjects was assessed by evaluation of

the incidence, nature, and severity of adverse events and serious adverse events (Supplementary Note 2). No radio-iodinated probe-related toxicity was evident clinically or by laboratory assay.

**Epichaperome selectivity**. The FP assay described above measures relative affinity constants without informing on epichaperome selectivity[40]. Because isolating the epichaperome for kinetic binding analyses in vitro is a technical challenge, a surrogate method is to monitor, after bolus administration, the clearance rate of PU-HZ151 from epichaperome-positive and -negative tissues (Fig. 3a)[13,21]. In essence, such experimental set-up mimics the classical method of measuring dissociation rate constants whereby an equilibrium mixture of ligand, receptor, and ligand-receptor complex are diluted and the time course of the dissociation of complex to establish new equilibrium concentrations of ligand and receptor, is observed.

We opted for the use of xenografted MDA-MB-468 and ASPC1 tumors in mice (target-positive versus target-negative, respectively, with equal total HSP90 levels[13], Fig. 3b). Because PU-HZ151 rapidly distributes throughout the body (see Fig. 2a and further below), and taking in consideration the high concentration of chaperones in cells (>100 μM)[41], a bolus injection creates a rapid equilibrium mixture state. Subsequent to reaching an 'equilibrium' state throughout the body, PU-HZ151 then rapidly clears from plasma, which initiates a body-wide dilution state. Recording the differential concentration-time curves for PU-HZ151 in the two tumor types offers the relative dissociation rate constants, but most importantly, informs on probe's epichaperome selectivity (Fig. 3a).

A probe that is selective for the epichaperome will clear slower from MDA-MB-468 than from ASPC1 tumors. It will also clear from normal tissue faster than from each tumor type, and with kinetics that resemble those of the blood pool[13,15,20,21]. This is the profile we observe for PU-HZ151 in mice, with clear retention and visualization of MDA-MB-468 tumors even at 48 h post-injection, a time when no signal was noted in ASPC1 tumors, plasma, or normal tissues (Fig. 3c, d), similar to what we reported for PU-H71 (Supplementary Fig. 17b and refs. [13,15]). The PET signal observed in the MDA-MB-468 tumor (i.e. target-expressing tissue) but not in the GI tract (i.e. clearance mechanism) corresponded to the intact PU-HZ151 molecule (Fig. 3e, f and Supplementary Fig. 16a–c).

From MDA-MB-468 tumors, PU-HZ151 cleared in a bi-exponential fashion. After an initial rapid clearance phase (0–6 h), a slow terminal clearance phase followed (24–96 h monitored) (Fig. 3c). The first phase reflects the clearance of blood-borne activity (see overlapping curves for tumors and heart, Fig. 3c) while the second is attributable to slow dissociation from the epichaperome target (see curves for MDA-MB-468 versus ASPC1, Fig. 3c). We corroborated these findings in human patients (Fig. 3g, h and Supplementary Note 2) where, as proof-of-principle, we demonstrated the retention of PU-HZ151 in a lung carcinoma, with clear differentiation from surrounding lung and muscle tissue at 3 h post-injection. We confirmed that intact PU-HZ151 was observed in both brain and epichaperome-positive tumors (Supplementary Fig. 16a–c), and that epichaperome-positive lesions and surrounding tissues were similarly perfused and accessible to the inhibitor (Fig. 3i). Combined, these studies verify that PU-HZ151 retains the kinetic preference of PU-H71 for the epichaperome over HSP90 pools[13,15].

Excretion of PU-HZ151 occurs via the hepatobiliary and urinary routes. For the radiolabeled PU-HZ151, the observed radioactivity in the GI tract is due to hepatic drug metabolism and hepatobiliary excretion (see GI clearance in Fig. 3d PET images). We also observed radioactivity in the thyroid which comes from the small fraction of free radioiodine released in vivo (see thyroid in Fig. 3d PET images). The release of a small amount of free radioiodine is not uncommon for radioiodinated probes, and in clinical practice, thyroid uptake is routinely and effectively blocked using oral administration of a saturated solution of potassium iodide prior to administration of such radio-agents[42,43].

**Epichaperome detection in brain tumors**. We next investigated if the CNS-directed epichaperome probes detect epichaperome positivity and engage the target in brain tumors. MDA-MB-468 is a triple-negative breast cancer cell line, an aggressive breast cancer subtype with high propensity of metastasizing to the brain. We therefore initiated our studies with intracranial MDA-MB-468 tumors. We imaged the real-time whole-body distribution of [124I]-labeled PU-HZ151 ([124I]-PU-HZ151) between 1 h and 72 h to observe that PU-HZ151 was evenly distributed throughout the body with brain exposure equaling that of other tissues and non-metabolizing organs (see PET image at 1 h, Fig. 4a). A rapid clearance from HSP90 (present throughout the brain) but not epichaperome (present only in the tumor), provided unambiguous visualization and quantitation of epichaperome positivity through PET imaging. No signal was observed in normal brain tissue or in other structures of the skull (Fig. 4b). We also confirmed target positivity, and specific target and probe co-localization, through radiography and associated hematoxylin and eosin staining of brain tissue (Fig. 4c–e).

**Epichaperomes are a glioblastoma hallmark**. Having established both the selectivity of our toolset for the epichaperome and its ability to detect it in the CNS, we proceeded to investigate its utility in the study of disease. We chose glioblastoma multiforme (GBM) a highly aggressive brain tumor with poor prognosis[44]. Whereas GBMs share common histological features, at a molecular level these tumors are highly variable from patient to patient and can display significant regional heterogeneity within the same tumor[45]. Discovery of mechanisms, and in turn targets that address such heterogeneity, as well as the development of non-invasive imaging techniques to detect such mechanisms (and targets) and monitor their dynamics during treatment, are of clinical significance and highly desirable.

We used several GBM tumors derived from glioma-stem cells (GSCs) stereotactically implanted into the mouse brain to evaluate the expression of the epichaperome, or the lack of, in GBMs (Fig. 5a–d). It is reported that GSC-initiated patient-derived xenografts mimic the features of the parental (i.e. patient's) tumor[46]. We observed epichaperomes are a characteristic of some GBMs, but not all, with epichaperome levels varying among the distinct GBMs we evaluated despite comparable total HSP90 levels across the GBM samples we evaluated (Supplementary Fig. 18). No epichaperome positivity was observed in the lesion-free regions of the brain (Fig. 4c–e and Fig. 5b–d, see H&E versus AR).

Epichaperome formation is a mechanism of tumor survival, and epichaperome expression is directly proportional to tumor's sensitivity to agents that induce its disassembly[13]. We therefore designed a testing paradigm where we measured both epichaperome expression and vulnerability of GBMs to PU-H71 and PU-HZ151[13]. We implemented into our study tumor samples from resective GBM surgery (Fig. 6a and Supplementary Table 11), where tumor cells are retained along GSCs, tumor endothelial

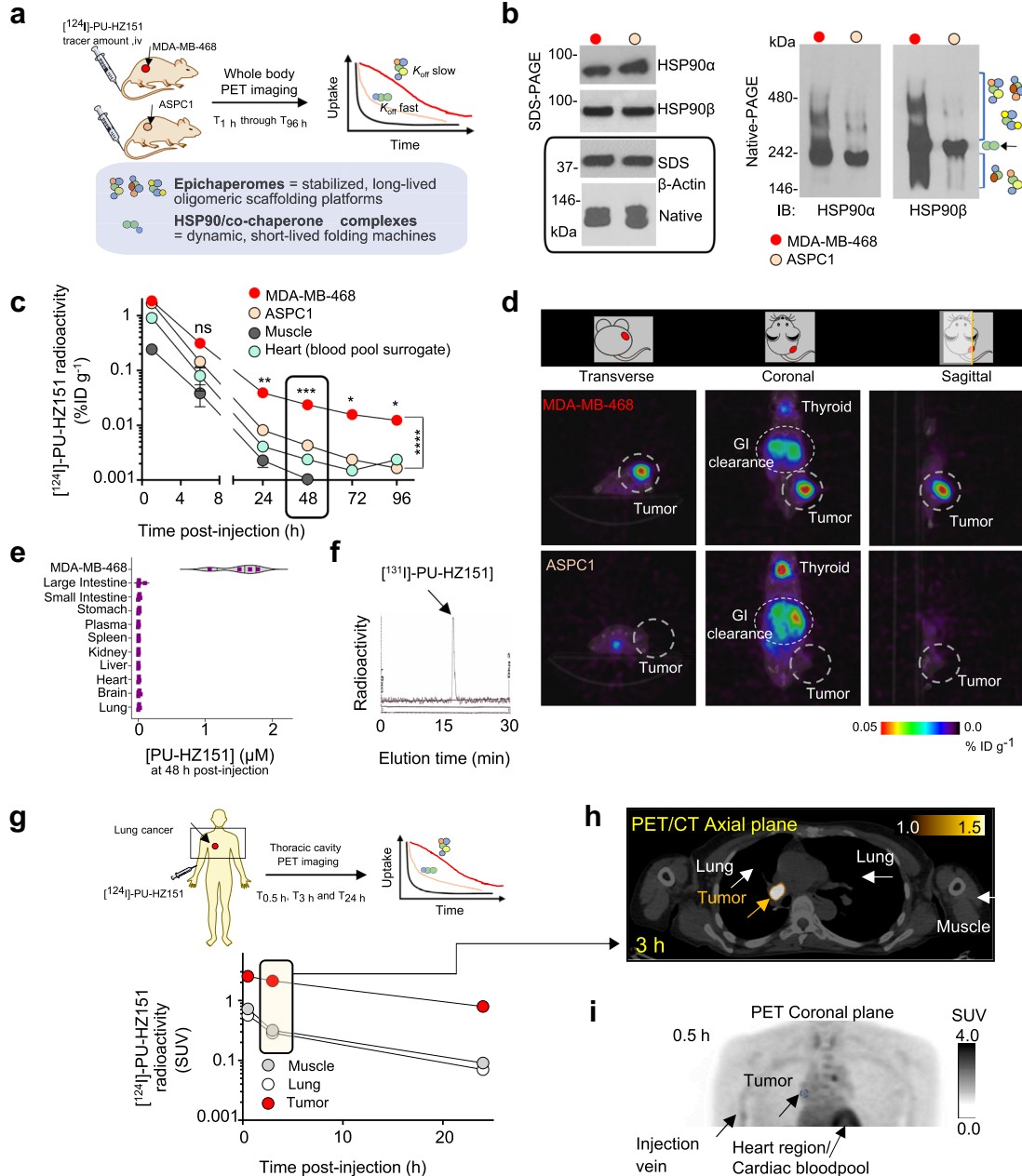

**Fig. 3 PU-HZ151 kinetically selects for the epichaperome over HSP90. a** Design of a surrogate method to determine epichaperome selectivity in vivo. iv, intravenous. **b** Native gels indicate epichaperome enrichment in MDA-MB-468 despite comparable total levels of HSP90 in both MDA-MB-468 and ASPC1 cells (as shown by denaturing gels). Epichaperomes appear as multiple high-molecular weight complexes in Native PAGE (see brackets). Conversely, HSP90 complexes, highly dynamic, and short lived, are detected as a band, putatively a homodimer (see arrow). β-Actin, loading control. **c** PET determined time-dependent concentration of $[^{124}I]$-PU-HZ151 in xenografted epichaperome +ve (MDA-MB-468) and −ve (ASPC1) tumors. Muscle and heart are shown for comparison. Serial PET images were obtained at the times indicated after single intravenous infusion of a microdose of $[^{124}I]$-PU-HZ151. Data are presented as mean ± s.e.m., two-way ANOVA with Sidak's post-hoc, $n = 5$ mice per tumor type. 24 h, $p < 0.0048$, 48 h, $p < 0.0002$, 72 h, $p < 0.0173$, 96 h, $p < 0.0120$; $F(1.009, 8.075) = 98.30$. **d** PET image of representative mice as in **c** obtained at 48 h post single $[^{124}I]$-PU-HZ151 injection (250 μCi; 12.5 μCi g$^{-1}$). **e** LC/MS-MS determined molar concentration of PU-HZ151 in the indicated tissues and organs at 48 h after single-dose intraperitoneal (i.p.) administration of 75 mg kg$^{-1}$ PU-HZ151. Graph, violin plot shows the distribution of individual mice with center line, median; box limits, upper and lower quartiles; whiskers, 1.5× interquartile range of $n = 4$. **f** Radio-HPLC of tumor extracts shows intact PU-HZ151 in MDA-MB-468 tumors (representative of $n = 4$) at 24 h post-single iv injection of 1 mCi $[^{131}I]$-PU-HZ151. See also Supplementary Fig. 16. **g** PET determined time-dependent concentration of $[^{124}I]$-PU-HZ15 (10 mCi) in a lung cancer patient. Serial PET images were obtained at the times indicated after single intravenous infusion of a microdose of $[^{124}I]$-PU-HZ151. **h** Representative axial PET/CT image obtained at 3 h as in **g**. **i** A 3-D maximum intensity projection (MIP) PET image of the chest, obtained at 0.5 h after iv injection of $[^{124}I]$-PU-HZ151 as in **g**, is shown to demonstrate the homogeneous distribution of the probe into tumor and surrounding tissues. Tumor's relative circumference is shown with a blue circle. Source data are provided as a Source Data file.

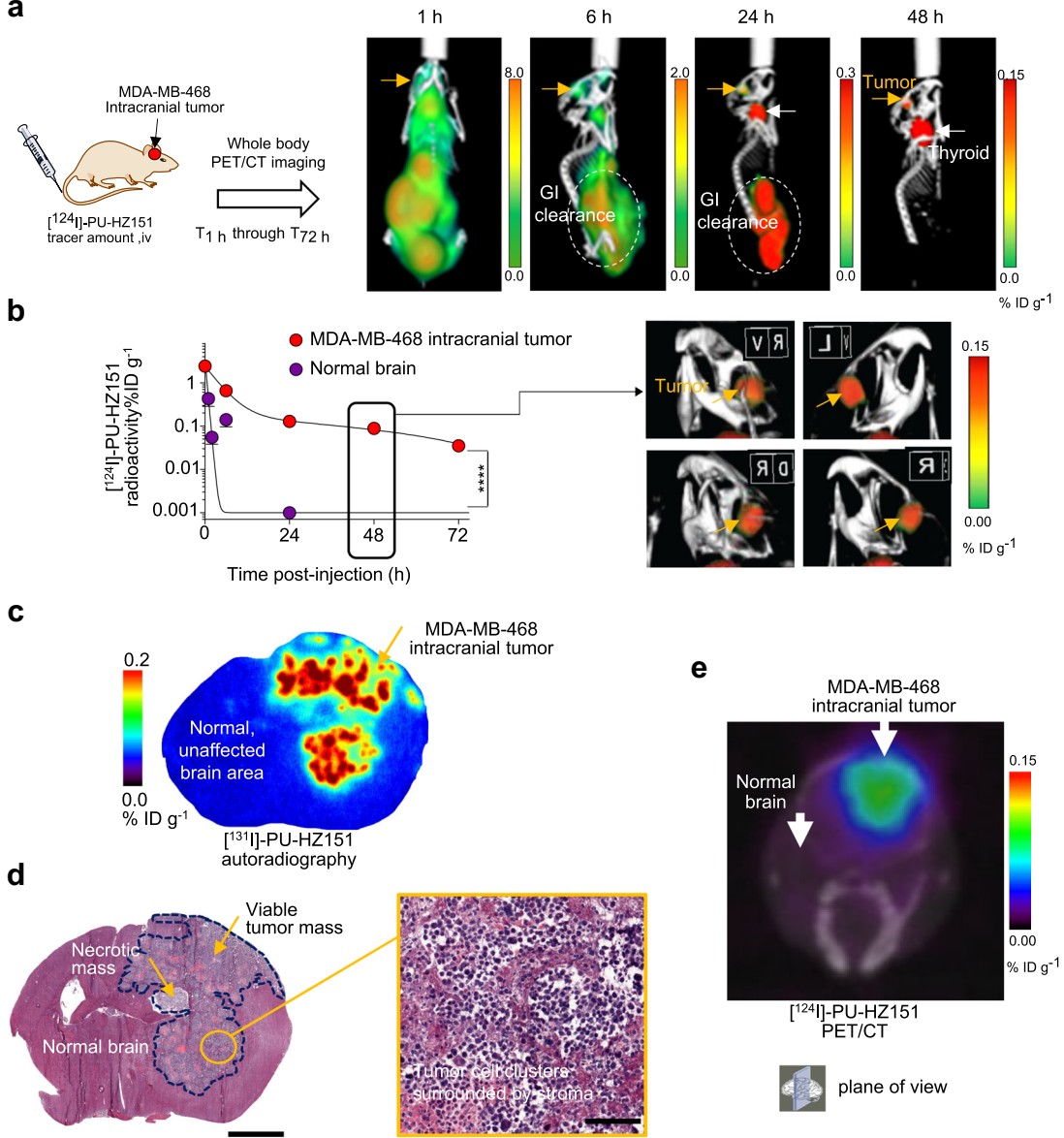

**Fig. 4 Radiolabeled PU-HZ151 detects epichaperome-positive lesions of the CNS. a** Experimental design to detect through PET imaging epichaperome-positive CNS lesions. Representative whole-body serial PET/CT images of mice as in **b** are also shown. **b** PET determined time-dependent concentration of [$^{124}$I]-PU-HZ151 in intracranial epichaperome+ve (MDA-MB-468) tumors and in the surrounding, unaffected brain. Serial PET/CT images were obtained at the times indicated after single intravenous infusion of a microdose of [$^{124}$I]-PU-HZ151 (15 μCi g$^{-1}$). PET/CT image of representative mice obtained at 48 h post-[$^{124}$I]-PU-HZ151 injection is also shown. Data, mean ± s.e.m., $n = 5$, two-way ANOVA, $p < 0.0001$, $F(1,40) = 91.62$. **c**, **d** Representative digital autoradiography ([$^{131}$I]-PU-AD, 15 μCi g$^{-1}$ iv, 6 h post-injection) (**c**) and H&E staining (**d**) of cryosections from brains as in **b**. The relative area occupied by the brain tumor is shown with a blue line. Inset in **d** shows the heterogeneity of the tumor with clusters of cancer cells surrounded by stroma. Scale bar, 2 mm. Inset scale bar, 100 μm. **e** PET scan of the mouse as in **c**, **d**, imaged and as in **b**, prior to killing. For (**b–e**) mice were first injected with [$^{124}$I]-PU-HZ151 and imaged up to 72 h. On the day following the last PET-scan, when [$^{124}$I]-PU-HZ151 was cleared, mice were injected [$^{131}$I]-PU-AD for autoradiography. The same sections were subsequently subjected to H&E staining for morphological evaluation of tissue pathology and to compare the localization of the epichaperome probe with the location of epichaperome-positive tumor tissue. Source data are provided as a Source Data file.

cells, and other tumor-associated cells[47]. Our goal was to investigate the sensitivity of these cells, and in turn their epichaperome dependence, in their native environment acknowledging that they function within an ecological system, both actively remodeling the microenvironment and receiving critical maintenance cues from their niches.

We prepared organotypic cultures from freshly resected tumor tissues ($n = 17$) and cultured them in media containing any of PU-H71, PU-HZ151, standard-of-care temozolomide (TMZ), and radiation (alone and combined), as allowed by the size of the available specimen. We included the anti-vascular endothelial

growth factor (VEGF) antibody Avastin (i.e. bevacizumab), approved for recurrent GBM. We observed that explants demonstrated vulnerability to the epichaperome probes despite resistance to approved therapies (Fig. 6b, ATP levels). Immuno-histochemical analysis confirmed apoptosis in both tumor parenchyma and in the tumor endothelium in explants treated with the epichaperome probes (Supplementary Fig. 19a, b TUNEL and caspase-3-positivity). We isolated tumor-associated endothelial cells from the explants and confirmed their sensitivity (Supplementary Fig. 19c–e). Lastly, we assessed explants for their ability to give rise to passageable neurospheres (NSs) and

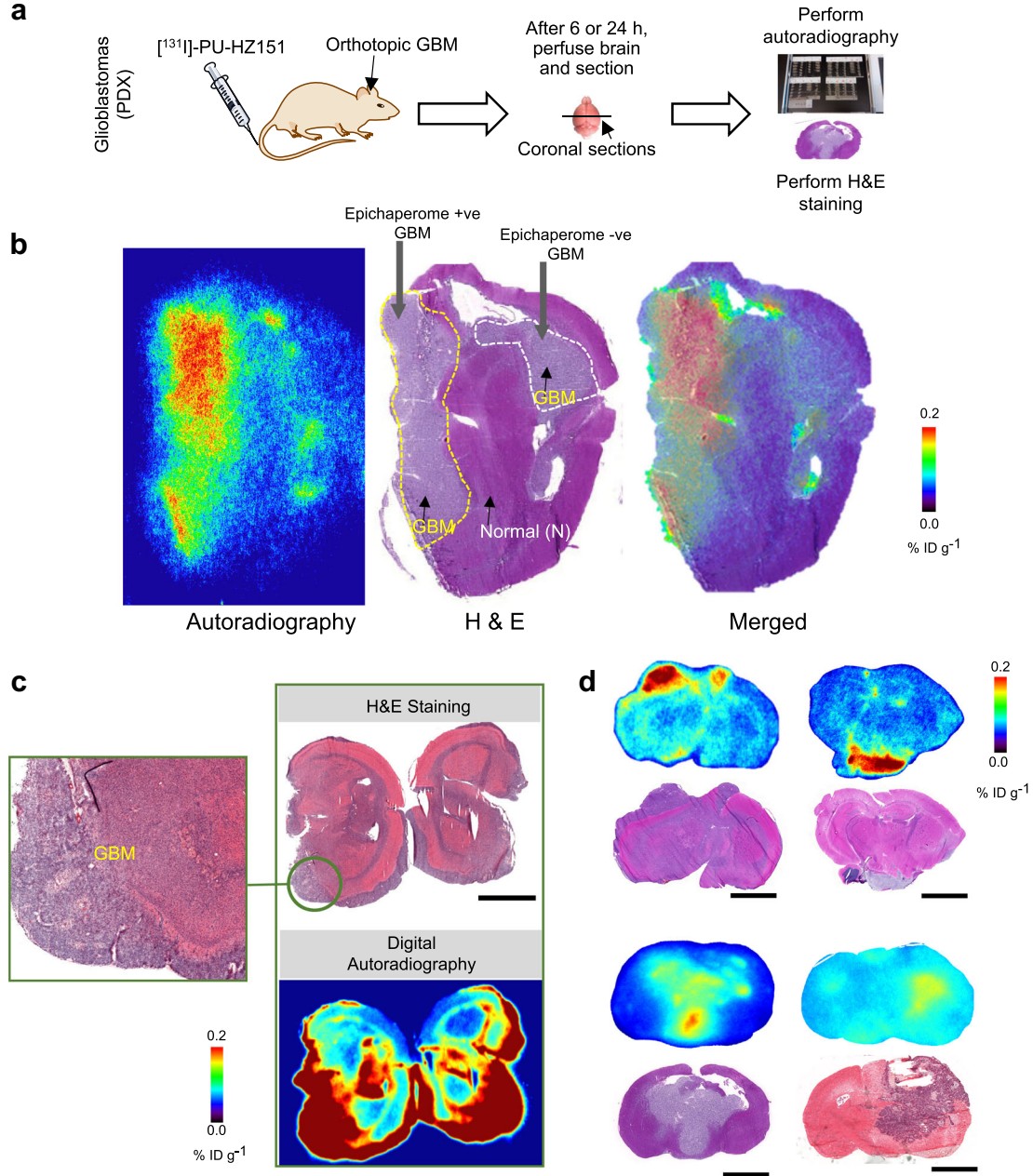

**Fig. 5 Radiolabeled PU-HZ151 detects epichaperome-positive glioblastoma patient-derived xenografted tumors. a** Experimental design to detect epichaperome-positive glioblastoma multiforme (GBM) lesions and differentiate them from those epichaperome-negative through autoradiography (AR). H&E was performed to differentiate the normal from the malignancy-afflicted brain regions. **b–d** Representative digital AR and H&E staining of brain cryosections. For **a**, **d**: 15 µCi g$^{-1}$ [$^{131}$I]-PU-HZ151 injected and AR performed at 6 h post-iv injection; for **c**: 35 µCi g$^{-1}$ [$^{131}$I]-PU-HZ151 injected and AR performed at 24 h post-iv injection. Each section is from an individual mouse. For **b**, both epichaperome-positive and -negative glioma-stem cells were stereotactically injected to form tumors. For **c**, inset shows a region with highly infiltrative GBM. Both epichaperome +ve and -ve GBMs have comparable HSP90 levels, see Fig. 6 and Supplementary Fig. 18. Scale bar, 2 mm.

identified five for which we also had data on their sensitivity to epichaperome probes (#2, #3, #5, #19, and #30). We observed NS eradication in 3 out of 5 explants, indicative of their epichaperome dependence (Fig. 6c). Vulnerability of GSCs to epichaperome probes and standard-of-care did not overlap, with TMZ and Avastin ineffective in specimen #30 that was vulnerable to epichaperome impairment (Fig. 6d). Thus, several cell populations that are important for GBM tumorigenesis, including tumor cells and tumor-associated cells, may form epichaperomes, and in turn, depend on epichaperomes for their survival, as we recently found in other malignancies[48].

While each population warrants further in-depth investigation, we focused initially on the GSCs, which are a source of tumor initiation. GSCs possess properties that make them resistant to both radiation and TMZ chemotherapy, the standard-of-care in GBM along with surgical resection. They also serve as a repository for GBM, leading inevitably to recurrence[49]. It is therefore believed that treatments targeting GSCs are of high significance in GBM and could extend patient survival, and possibly result in cures[50]. For several GBMs, and in addition to those evaluated at the explant level, we were able to establish GSC cell lines. We used these cultured GSCs to profile each specimen

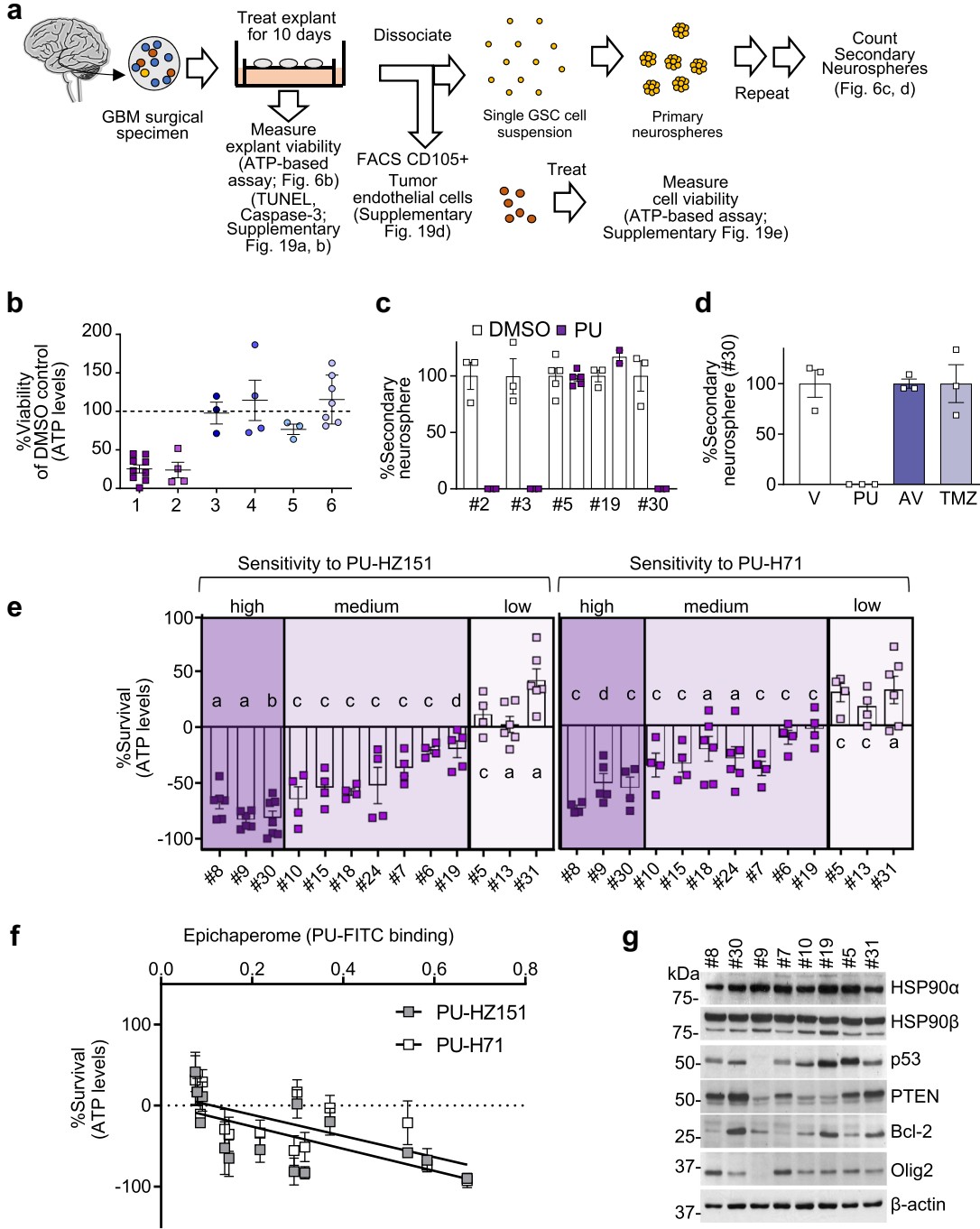

for epichaperome levels and for molecular characteristics, in addition to vulnerability ($n = 13$). We observed a spectrum of vulnerabilities of these cells to epichaperome inhibition, which is in line with what we observed in the explants and also for other tumor types[13]. Some GSCs were highly sensitive ($n = 3$, 80–100% cell death of the initial cell population), some exhibited a moderate sensitivity level ($n = 7$), whereas others were resistant ($n = 3$). GSCs sensitive to PU-H71 were also those sensitive to PU-HZ151, supporting, functionally, their target commonality (Fig. 6e).

Sensitivity to both PU-H71 and PU-HZ151 appeared to correlate with epichaperome status, with most sensitive samples exhibiting greater epichaperome levels (Fig. 6f, $r^2 = 0.4$, $p < 0.0001$). The total levels of HSP90 and of other chaperones

involved in epichaperome formation (eg. HSC70, HSP70) were comparable between the different GBM specimens we assessed for sensitivity (Fig. 6g and Supplementary Fig. 18). There was no clear relationship in this study between sensitivity to epichaperome impairment and common molecular features of GBM[51], expression of pro- and anti-apoptotic markers or HSP90 client proteins, a hallmark of drug action via HSP90[52].

**Epichaperome imaging for GBM stratification.** To test if our probes can detect and differentiate in vivo the epichaperome-dependent tumors from those that are not, we next performed intracranial xenotransplantation of GSCs from epichaperome-high (GSC#8), -medium (GSC#7; GSC#19) and -negative

**Fig. 6 Human GBM express variable levels of epichaperomes, with epichaperome-positivity portending vulnerability to agents that induce epichaperome disassembly. a** Schematic representation of explant treatment, viability assessment and serial neurosphere (NS) formation assay. Cytotoxicity of agents on GBM explants was determined through gross evaluation of ATP levels and also by co-staining of explants with markers of cell identity and cell death. Explant tissue was also dissociated to obtain a single cell suspension. The self-renewal capacity of the glioma-stem cells (GSCs) was evaluated by performing serial NS formation assays. Lastly, dissociated explant tissue was sorted by FACS to isolate tumor endothelial cells, which we then cultured in the presence or absence of the agents. Surgically derived human GBM explants were treated for 10 days, and then were dissociated into single cell suspensions to analyze the self-renewal capacity of the cells. Treatments include: PU-HZ151 (1 μM), PU-H71 (1 μM), Avastin (AV) (1 μM), temozolomide (TMZ) (250 nM), a single dose of radiation (XRT) (10 Gy on day 5) or DMSO, and control vehicle (V). **b** Viability of GBM explants as in **a** to 1, PU-H71 ($n = 9$); 2, PU-HZ151 ($n = 4$); 3, AV ($n = 3$); 4, TMZ ($n = 4$); 5, XRT ($n = 3$); and 6, TMZ + XRT ($n = 7$). $n$, individual GBMs as in **a**. See also Supplementary Fig. 19. **c**, **d** Secondary neurosphere formation as in **a**. #2, #3, #19, #30, $n = 3$ and #5, $n = 5$ biological replicates from independent NSs of individual patients. Both PU-H71 and PU-HZ151 were used as in **a**, with similar results. **e** Cell viability assessed by an ATP-based assay in patient-derived GSCs cultured adherently in a serum-free stem cell culture condition and incubated for 48 h with PU-HZ151 or PU-H71 (1 μM). Negative values represent a decrease of cell numbers compared to the initial cell population. Each bar represents an individual GSC, a, $n = 6$; b, $n = 7$; c, n = 4; d, n = 5 experimental repeats. **f** Correlative analysis between cytotoxicity evaluated as in **e** and epichaperome levels evaluated by flow cytometry (see methods). Pearson's $r$, two-tailed, $n = 13$ individual GSCs as in **e**. **g** Western blot analysis of molecular markers characterizing each GSC. β-actin, loading control. See also Supplementary Fig. 18. All graphs, data are presented as mean ± s.e.m. Source data are provided as a Source Data file.

(GSC#5) specimens. For comparison we used intracranially xenotransplanted MDA-MB-468 tumors (Fig. 7a).

We performed PET imaging using [$^{124}$I]-labeled PU-HZ151 to evaluate the epichaperome status of the GSC-derived GBM tumors (Fig. 7b, see AUC$^{tumor/blood}$ and AUC$^{brain\ tumor/normal\ brain}$). We observed that GBM#7, GBM#19 and GBM#5 retained the molecular feature of the parental GSC, in that GBM#7 had epichaperome levels closer to the MDA-MB-468 tumors (AUC$^{tumor/blood}$ = 893 and 1303, respectively), GBM#5 was low to negative (AUC$^{tumor/blood}$ = 143), whereas GBM#19 placed in between (AUC$^{tumor/blood}$ = 444).

Interestingly, GBM#8, which was a high-epichaperome/high-sensitivity GSC in culture, became an epichaperome-negative GBM tumor upon transplantation (Fig. 7b, AUC$^{tumor/blood}$ = 80). We confirmed this change in molecular features by analyzing both GSC#8 cells and GBM#8 tumors by native gels, western blot (Fig. 7c and Supplementary Fig. 20) and [$^{131}$I]-PU-HZ151 autoradiography (Fig. 7d). In GSC#8, we observed a number of distinct and indistinct high-molecular weight HSP90 species which as reported, is the biochemical signature of stable, oligomeric HSP90 incorporated into epichaperomes[13,14]. In GBM#8, Native PAGE captured mainly the HSP90 dimer and monomer, supportive of epichaperome-loss and the switch to a tumor mechanism that is epichaperome independent. The parental tumor and the GSC#8 population were characterized by EGFR overexpression (see Supplementary Table 10, EGFR amplification and Supplementary Figs. 18 and 20, high EGFR levels on western blot), a feature also lost in the GBM#8.

There was a positive correlation between epichaperome expression determined by [$^{124}$I]-PU-HZ151 PET and the vulnerability of these GSCs as determined in culture (for PU-HZ151 $r^2 = 0.8697$, $p < 0.0001$; for PU-H71 $r^2 = 0.8812$, $p < 0.0001$) (Fig. 7d). This was also reflected in their sensitivity to epichaperome inhibitor treatment in vivo (Fig. 7e). We observed that PU-HZ151 significantly improved the survival of epichaperome-positive GBM#7 ($\chi^2 = 7.51$, $P = 0.0061$) and MDA-MB-468 ($\chi^2 = 14.51$, $P = 0.0001$) but not of epichaperome-negative GBM#5 ($\chi^2 = 0.03$, $P = 0.86$) and GBM#8 ($\chi^2 = 1.57$, $P = 0.21$) bearing mice.

These findings combined, are confirmatory of the probes' selectivity for the epichaperome over HSP90 and their ability to detect, engage, and modulate epichaperomes in the CNS (Fig. 8).

## Discussion

Discovery of CNS-directed epichaperome probes is encumbered by a lack of clear information on the structure and conformation adapted by HSP90 and other chaperome members in the context of the epichaperome. To overcome this limitation, our study makes use of HSP90 pocket architecture captured by PU-H71 to design ligands with properties that may improve brain delivery. It then implements a battery of tests designed to probe epichaperome engagement, specificity, and selectivity at the cellular and organismal level, including mice and humans.

In addition to probes, our study introduces methods and protocols for the use of the toolset in cellular models as well as in mice, with proof-of-principle provided for their use in human patients. We show that detection and quantitation of epichaperomes is achievable with these agents and our protocols through PET imaging or through radiography.

Our extensive data clearly demonstrate the molecular specificity of the probe. There are three key parameters that combined determine brain uptake and target engagement in the CNS: 1. the PK and the exposure of the brain lesions to the chemical probe (to understand brain delivery); 2. the ratio of probe's presence in target-positive lesions over normal brain and target-negative lesions (to understand selectivity and specificity, i.e. no binding to target-negative brain structures) and 3. the functional engagement of the target-positive lesions over the target-negative lesions (to demonstrate productive target engagement). Though uptake determines technically if a compound enters brain tissue, this parameter alone is inadequate to determine if such brain penetration results in ligand–target interaction[53]. Our strategy covers all three aspects, to show desirable PK, specific exposure of the target, and functional engagement of the target in the target-positive lesions over the normal brain and target-negative lesions.

In solid tumors, the enhanced permeability and retention (EPR) effect could confound nonspecific vascular leakage and specific binding[54]. Although EPR mostly concerns high-molecular weight compounds and not small molecules such as PU-HZ151, we used in our study both target-positive and -negative tumors in our animal models to exclude the possible implication of the EPR in the observed retention of the probe in epichaperome-positive lesions. This is in addition to a multitude of biochemical, analytical, and functional assays in vitro and in vivo, including proof-of-principle imaging studies in humans, that conclusively demonstrate retention of the probe in target-expressing brain lesions with minimal/no retention in the non-target-expressing brain lesions and normal brain.

Our study also has important therapeutic significance that cannot be underestimated in terms of its treatment potential for multiple, diverse disease states in the cancer field and within the CNS. Our findings propose epichaperome formation is a mechanism for the survival of some GBMs and GBM-associated cell populations (eg. GSCs and tumor endothelial cells), a

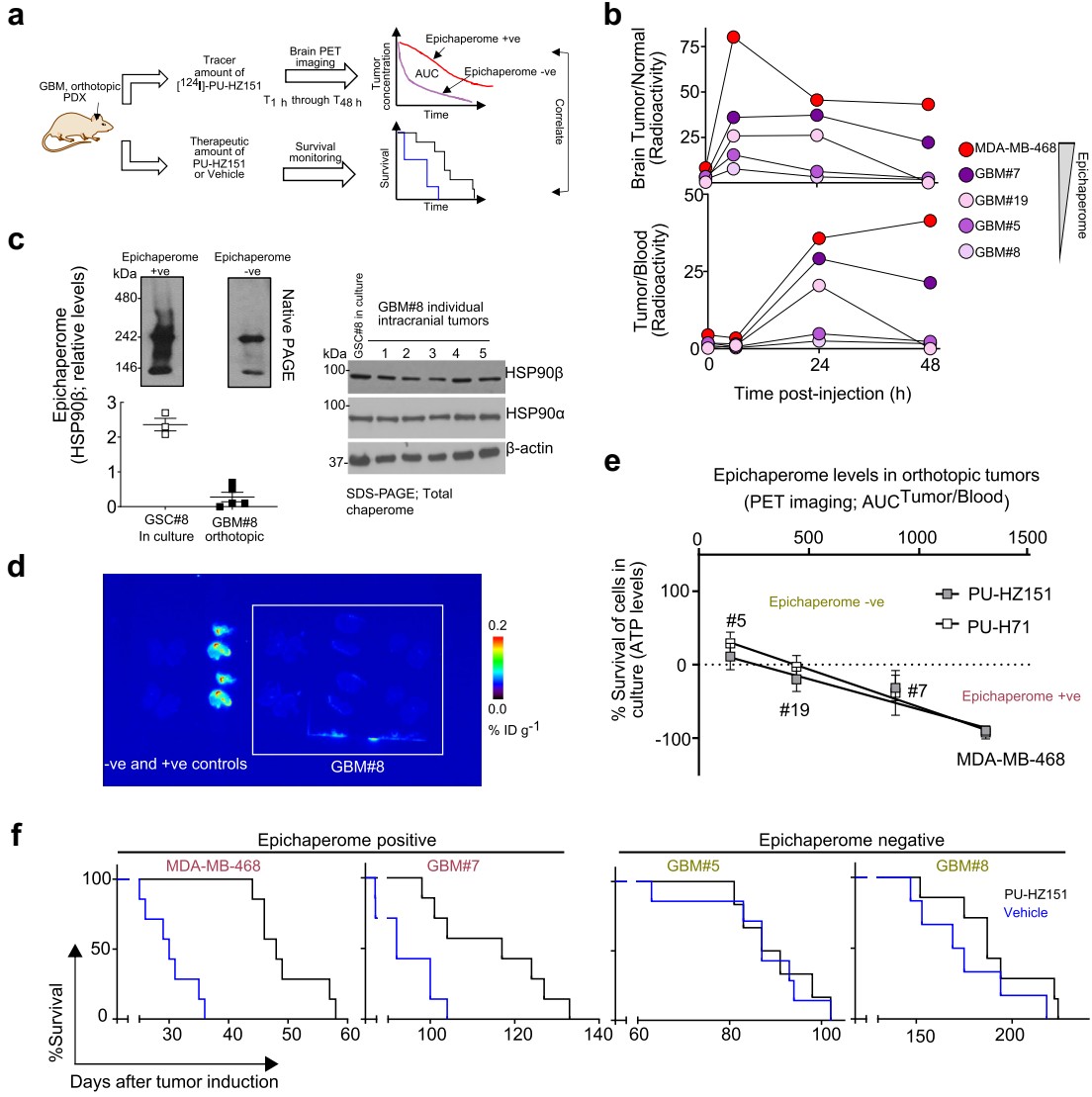

**Fig. 7 PET imaging with [$^{124}$I]-PU-HZ151 detects and quantifies epichaperome-positivity. a** Schematic of the experimental design. PDX patient-derived xenograft. **b** PET determined time-dependent concentration of [$^{124}$I]-PU-HZ151 in intracranial tumors, in the surrounding, unaffected brain, and in the plasma. Serial PET/CT images were obtained at the times indicated after single intravenous infusion of a microdose of [$^{124}$I]-PU-HZ151 (15 µCi g$^{-1}$). Data are plotted as ratio of the recorded PET signals. $n = 5$ mice per tumor type. **c** Epichaperome quantification in the GBM#8 tumors ($n = 5$) from (**b**) compared to levels recorded in cultured GSC#8 cells ($n = 3$ cultures). Data are presented as mean ± s.e.m. Representative Native PAGE immunoblotted for HSP90 in epichaperomes and Western blot analyses of total HSP90 levels are also shown. See also Supplementary Fig. 20. **d** [$^{131}$I]-PU-HZ151 autoradiography of individual GBM#8 PDX tumors ($n = 3$), 4 explants per PDX, as in **c** confirms loss of epichaperome positivity in GBM#8 upon xenotransplantation. **e** Correlative analysis between GSC cytotoxicity to PU-HZ151 and PU-H71, evaluated as in Fig. 6, and epichaperome levels, evaluated by [$^{124}$I]-PU-HZ151 PET. Pearson's $r$, two-tailed, $n = 4$ tumor types, as indicated. **f** Kaplan–Meyer survival curves for mice bearing the indicated intracranial tumors. Vehicle, blue curve; PU-HZ151 (50 mg kg$^{-1}$), black curve. $n = 7$ mice per condition. Source data are provided as a Source Data file.

mechanism that is independent of the histologic and molecular complexity observed in GBMs. Our data show these GBMs are detectable through PET imaging using the [$^{124}$I]-PU-HZ151 probe and actionable using the PU-HZ151 probe, which is also a drug candidate. Both PU-HZ151 and [$^{124}$I]-PU-HZ151, under the name of PU-AD and PU-AD PET, respectively, are now in clinical development as potential treatment and companion diagnostic, in AD patients. PU-AD has finalized Phase 1 in healthy patients (NCT03935568) and is now in Phase 2 clinical evaluation in patients with mild AD (NCT04311515), whereas PU-AD PET has completed a pilot proof-of-principle study (NCT03371420). In AD, epichaperomes negatively impact assembly of proteins, which creates dysfunction of interactome networks implicated in AD-related processes and biological

functions known to decline in AD. These include key networks: synaptic plasticity, cell-to-cell communication, protein translation, axon guidance, metabolic processes, cell cycle, and inflammation[16]. With the radiolabeled [$^{124}$I]-PU-HZ151 probe now in clinic in the context of PET imaging, and the drug candidate PU-HZ151 in use for disease treatment, their rapid dissemination in the context of GBM is feasible.

Because of their kinetic selectivity for the epichaperome over the abundant HSP90 chaperone proteins, PU-HZ151 follows a hit and run profile in normal tissue (where HSP90 resides) but a hit and stay profile in the epichaperome-positive diseased tissue. In this paradigm, PU-HZ151 rapidly distributes to the site of disease – the hit stage – but also rapidly clears from normal tissue – the run stage, while engaging the target in the diseased cell for days

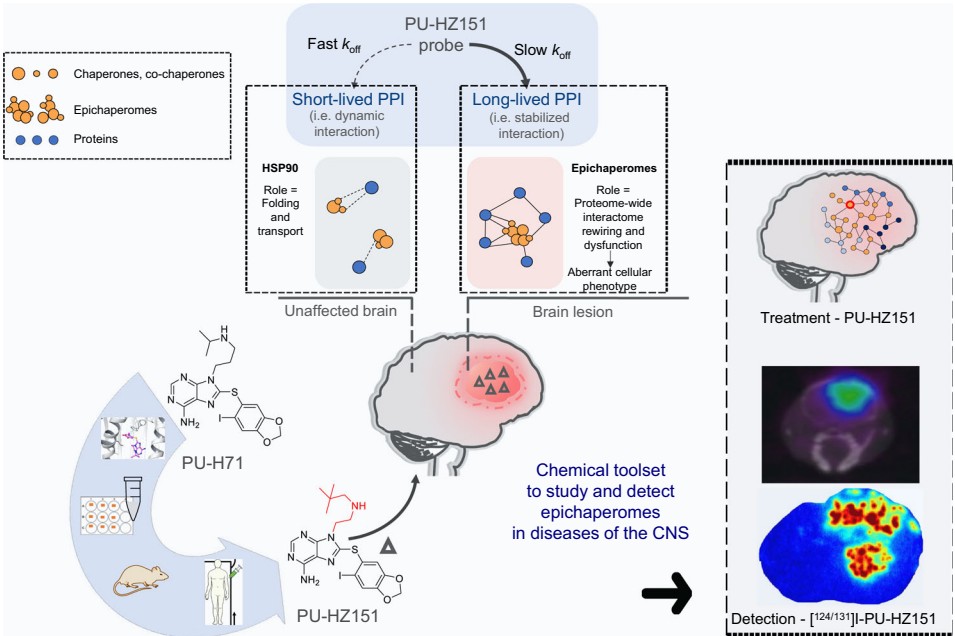

**Fig. 8 Discovery platform for brain-directed chemical probes that kinetically select for epichaperomes over the more abundant HSP90.** We here introduce a platform composed of structural, biochemical, and functional insights for discovery of BBB-permeable epichaperome probes and demonstrate its success in the discovery of a CNS-directed chemical toolset. We show how the platform yields a toolset consisting of probes to study, detect, and treat epichaperome-mediated interactome network dysfunctions in cells, mice, and humans.

after single drug administration – the stay stage. Thus, when mixed live with all the proteins present in the mouse body, and in spite of the large abundance of HSP90 – a protein found essentially in all mammalian cells and estimated to be ~224–336 g in a mature human weighting 70 kg and estimating a 16% protein mass – PU-HZ151 specifically and with a long residence time engages the epichaperome target present in the diseased brain. Thus, due to the precise nature of epichaperome expression in the diseased cells and PU-HZ151's long target binding half-life, PU-HZ151 can have a duration of action which extends beyond its presence in plasma, thereby limiting potential activity at off-target proteins. Using this strategy, epichaperome targeting in cancer by PU-H71 has shown both safety and efficacy in historically difficult to treat cancers such as triple-negative breast cancer and myeloproliferative neoplasm transformed to acute myeloid leukemia[48,55].

While genetic, transcriptomic and proteomic profiling of GSC is proposed as a modality for the discovery of targets for this important cell fraction, we use our probes to uncover a functional signature – epichaperome-mediated proteome connectivity dysfunction (i.e. interactome network dysfunctions induced by PPI perturbations)[1] – as a potential target with diagnostic and therapeutic upside. We show that GSCs exhibiting this hallmark are identifiable and targetable using the probes we discover here. GSCs are maintained within a special microenvironment, termed niche, which regulates their activity and cell-fate decision. Among them, tumor endothelial cells impact the biology of GSCs by both direct interaction and through the production of cytokines, such as VEGF. By confirming that both tumor endothelial cells and GSCs require epichaperomes to survive, we uncover that cells in the microenvironment, in addition to tumor cells, are impacted by this mechanism. Epichaperomes may therefore provide a strategy for targeting both the glioma cells and the perivascular niche simultaneously. These hypotheses, on mechanisms and targetability, can be investigated with the toolset we introduce here.

In conclusion, we provide a comprehensive report on a discovery platform for CNS-directed epichaperome probes. Data herein provides proof-of-principle applications in the use of the toolset in the detection, quantification, and modulation of the target in complex biological systems such as the CNS, and lastly it provides evidence in the power of the toolset in uncovering biological and therapeutic knowledge in the context of human disease.

## Methods

**Reagents and chemical synthesis.** All commercial chemicals and solvents were purchased from Sigma Aldrich or Fisher Scientific and used without further purification. The identity and purity of each product was characterized by MS, HPLC, TLC, and NMR. Purity of target compounds has been determined to be >95% by LC/MS on a Waters Autopurification system with PDA, MicroMass ZQ, and ELSD detector and a reversed phase column (Waters X-Bridge C18, 4.6 × 150 mm, 5 μm) eluted with water/acetonitrile gradients, containing 0.1% TFA. Stock solutions of all inhibitors were prepared in molecular biology grade DMSO (Sigma Aldrich) at 1000× concentrations. New materials reported here are available from the authors on request. For detailed synthetic procedures, see Supplementary Notes.

**Human subjects.** Surgical glioblastoma specimens were obtained in accordance with the guidelines and approval of the Institutional Review Board# 09–156, PI: Viviane Tabar, MD. The patients provided signed informed consent prior to participation. Samples were de-identified before receipt for use in these approved studies.

**Trial design and patient population.** This first in-human trial of the positron-emitting agent [124I]-PU-HZ151 (124I-PU-AD, clinical name) was an open label pilot microdose PET-CT study (Dunphy, M. PET Imaging of Subjects Using 124I-PU-AD available from: http://clinicaltrials.gov; NCT03371420). The study was approved by the institutional review board (protocol 16-004), and conducted under an exploratory investigational new drug (IND) application approved by the US Food and Drug Administration. Patients provided signed informed consent before participation. The study was conducted in accordance with the Declaration of Helsinki. Adult patients with solid malignancy, lymphoma, or Alzheimer's disease were eligible (eligibility criteria detailed in Supplementary Table 12). The trial was first registered on 04.19.2016 as an investigator-initiated trial and started accruing at Memorial Sloan Kettering Cancer Center between 09.21.2016 and 07.21.2017. On 12.13.2017 a second registration on clinicaltrials.gov reflects this protocol IND being transferred over to Samus Therapeutics Inc. Five patients were enrolled, out of which three patients participated in the trial. Two additional patients withdrew consent as technical and logistical issues prevented participation. No hypotheses were statistically tested in this small pilot clinical study[56]. For thyroid protection against possible free radioiodine, patients ingested seven drops of saturated solution of potassium iodide (SSKI) at least 2 h prior to radiotracer injection repeating

SSKI dose daily for 2 weeks post-injection. To detect thyroid injury, serum thyroid-stimulating hormone (TSH) assay was obtained <1-week pre-injection, as baseline; with follow-up TSH assay 6–12 months post-injection. Patients were monitored for signs and symptoms of toxicity at each imaging time point. Patients reported any adverse symptoms experienced in the following 30 days to study investigators. Primary outcome measures were: Pharmacokinetic profile of 124I-PU-AD: area under the curve (AUC) [Time Frame: 1 week]; Pharmacokinetic profile of 124I-PU-AD: maximum plasma concentration ($C_{max}$) [Time Frame: 1 week]; Pharmacokinetic profile of 124I-PU-AD: trough plasma concentration ($C_{min}$) [Time Frame: 1 week]; Pharmacokinetic profile of 124I-PU-AD: plasma half-life ($T_{1/2}$) [Time Frame: 1 week]; Pharmacokinetic profile of 124I-PU-AD: time to maximum plasma concentration ($T_{max}$) [Time Frame: 1 week]. The results are reported in Supplementary Table 10. Secondary outcome measures were: incidence of adverse events [Time Frame: 30 days]. The [124I]-PU-HZ151 probe was synthesized in-house by the institutional cyclotron core facility at high specific activity. For the PET study, two intravenous catheters (heparin-locked) were placed in the subject for radiopharmaceutical administration and for blood sampling. After [124I]-PU-HZ151 tracer injection, PET scans were performed at the following time-points: immediately post-injection; 2.5–4.5 (±30 mins) h post-injection; and 1–2 days post-injection. Optionally, in willing patients, an additional PET scan was obtained 3–7 days post-injection. At each time-point, a 30–60 min axial body image (spanning from skull vertex to proximal thigh regions) was acquired on a state-of-the-art PET-CT scanner. A low-dose CT was obtained immediately-prior to PET imaging, at each timepoint. A 30–60 min scanning time-period is typical for clinical nuclear medicine diagnostic imaging studies. Subjects were monitored visually, and communication maintained directly between the patient and the radiology investigators, except during the CT, when communication is maintained via a speaker system. The radiology investigator evaluated patients during the early time-period, post-injection. See section below Epichaperome detection in human patients. Serial blood samples were obtained at approximately 1, 5, 15 ± 5, 30 ± 5, 60–90 (±30) minutes; and 2.5–4.5 (±30 mins) hours, post-injection. See section below on Plasma pharmacokinetics assays in human patients. Subjects were evaluated to ensure that there are no clinically significant ongoing adverse effects prior to discharge. The microdose study NCT0126959339, PET Imaging of Cancer Patients Using 124I-PUH71; A pilot study, was conducted as previously reported[39].

**Mouse models**. All animal studies were conducted in compliance with MSKCC's guidelines and under Institutional Animal Care and Use Committee (IACUC) approved protocols #05-11-024 and #04-03-009. Athymic nude mice (Hsd:Athymic Nude-Foxn1nu, female, 20–25 g, 6 weeks old; RRID:MGI:5652489) and B6D2F1 mice (male, 4–5 weeks, Jackson Laboratory; RRID:IMSR_JAX:100006) were allowed to acclimatize at the MSKCC vivarium for 1 week prior to experiments. Mice were housed in groups of 4–5 mice per individually ventilated cage in a 12-h light/dark cycle (6:00 a.m./6:00 p.m.), with controlled room temperature (22 ± 1 °C) and humidity (30–70%). Mice were provided with food and water ad libitum. All mice in all studies were observed for clinical signs at least once daily.

**Cell lines and culture conditions**. The MDA-MB-468 (HTB-132; RRID: CVCL_0419), Kasumi-1 (CRL-2724; RRID:CVCL_0589), HepG2 (HB-8056; RRID: CVCL_0027), ASPC1 (CRL-1682; RRID:CVCL_0512), and HEK-193 (CRL-1573; RRID:CVCL_0045) human cancer cell lines were obtained from the American Type Culture Collection and cultured in Dulbecco's Modified Eagle's medium-high glucose (DME-HG) supplemented with 10% FBS, 1% L-glutamine, 1% penicillin, and streptomycin. Cells were authenticated using short tandem repeat profiling and tested for mycoplasma.

**Fluorescence polarization**. For the binding studies, fluorescence polarization (FP) assays were performed similarly as was previously reported[37]. Briefly, FP measurements were performed on an Analyst GT instrument (Molecular Devices, Sunnyvale, CA). Measurements were taken in black 96-well microtiter plates (Corning # 3650) where both the excitation and the emission occurred from the top of the well. A stock of 10 µM cy3B-GM was prepared in DMSO and diluted with HFB buffer (20 mM Hepes (K), pH 7.3, 50 mM KCl, 2 mM DTT, 5 mM MgCl₂, 20 mM Na₂MoO₄, and 0.01% NP40 with 0.1 mg mL⁻¹ BGG). The test compounds were dissolved in DMSO and added at several concentrations to the HFB assay buffer containing both 6 nM cy3B-GM and mouse brain lysate (6 µg JNPL3 mouse brain lysate) in a final volume of 100 µL. Drugs were added to triplicate wells. Free cy3B-GM (6 nM cy3B-GM), bound cy3B-GM (6 nM cy3B-GM + lysate, as indicated above), and buffer only containing wells (background) were included as controls in each plate. Plates were incubated on a shaker at 4 °C, and polarization values measured at 24 h. Percentage inhibition was calculated as follows: (% Control) = 100−(($mP_c$ − $mP_f$)/($mP_b$ − $mP_f$)) × 100, where $mP_c$ is the recorded mP from compound wells, $mP_f$ is the average recorded mP from cy3B-GM–only wells, and $mP_b$ is the average recorded mP from wells containing both cy3B-GM and lysate, and plotted against values of competitor concentrations. The inhibitor concentration at which 50% of bound cy3B-GM was displaced was obtained by fitting the data using a nonlinear regression analysis as implemented in Prism 7.0 (Graphpad Software).

**Competitive binding assay**. A stock of 10 µM PU-FITC[57] was prepared in Felts buffer (20 mM Hepes (K), pH 7.3, 50 mM KCl, 2 mM DTT, 5 mM MgCl₂, 20 mM Na₂MoO₄, and 0.01% NP40). In all, 200 µg of protein lysates prepared from the MDA-MB-468 cells were added into 96-well microplates (Greiner Microlon Fluotrac 200) and treated with vehicle or PU-H71 and PU-HZ151 (1 µM) at room temperature for 10 min. PU-FITC (10 nM) was added to each well in a final volume of 100 µL Felts buffer. To account for background signal, buffer and PU-FITC only controls were included in each assay. The FP values in mP were measured every 5–10 min. The assay window was calculated as the difference between the FP value recorded for the bound fluorescent tracer and the FP value recorded for the free fluorescent tracer (defined as mP − mPf). Measurements were performed on a Molecular Devices SpectraMax Paradigm instrument (Molecular Devices, Sunnyvale, CA), and data were imported into SoftMaxPro6 and analyzed in GraphPad Prism 7.

**General side effect profile II diversity panel**. PU-HZ151's binding to each of the 70 key receptor, enzyme, and ion channel proteins comprising the diversity panel screen (General SEP II) was performed by Caliper Life Sciences (now PerkinElmer). Binding was expressed as a mean percent of the reference control ($n = 2$ measurements) collected at a single, high (10 µM) concentration of PU-HZ151. NOVASCREEN suggests these guidelines for interpretation of the data presented: Baseline, −20% to +20% inhibition: In most assays, the standard baseline range runs from −20% to +20% inhibition of binding or enzyme activity. NOVASCREEN considers compounds showing results in this range inactive at this site. Compounds which show negative inhibition (< 20%): NOVASCREEN's assays are designed to test for inhibition of binding or enzyme activity. Occasionally, compounds will demonstrate high negative inhibition (i.e., resulting from the extraction procedure used) and may, at the discretion of the client, warrant retesting at lower concentrations. Compounds which show inhibition in the range of 20–49%: Compounds exhibiting these results show marginal activity at the receptor site and generally do not warrant further examination unless otherwise directed by the client.

**scanEDGE off-target screen**. Because PU-type ligands are based on a purine scaffold, there is the misconception that they may nonspecifically interact with ATP-binding pockets, such as those of kinases. PU-HZ151 was tested at 10 µM against the scanEDGE 97 kinase panel. This panel contains a set of kinases covering AGC, CAMK, CMGC, CK1, STE, TK, TKL, lipid, and atypical kinase families, plus important mutant forms. Developed by Ambit Biosciences, it employs proprietary active-site-dependent competition binding assays to determine how compounds bind to kinases. It is based on a competition binding assay that quantitatively measures the ability of a compound to compete with an immobilized, active-site-directed ligand and can be used in detection of multiple inhibitor types (e.g., type I, type II, and non-ATP competitive). For most assays, kinase-tagged T7 phage strains were grown in parallel in 24-well blocks in an *Escherichia coli* host derived from the BL21 strain. *E. coli* cells were grown to log-phase and infected with T7 phage from a frozen stock (multiplicity of infection = 0.4) and incubated with shaking at 32 °C until lysis (90–150 min). The lysates were centrifuged (6000×g) and filtered (0.2 µm) to remove cell debris. The remaining kinases were produced in HEK-293 cells and subsequently tagged with DNA for qPCR detection. Streptavidin-coated magnetic beads were treated with biotinylated small molecule ligands for 30 min at room temperature to generate affinity resins for kinase assays. The liganded beads were blocked with excess biotin and washed with blocking buffer [SeaBlock (Pierce), 1% BSA, 0.05% Tween 20, 1 mM DTT] to remove unbound ligand and to reduce nonspecific phage binding. Binding reactions were assembled by combining kinases, liganded affinity beads, and test compounds in 1× binding buffer (20% SeaBlock, 0.17× PBS, 0.05% Tween 20, 6 mM DTT). Test compounds were prepared as 40× stocks in 100% DMSO and directly diluted into the assay. All reactions were performed in polypropylene 384-well plates in a final volume of 0.04 mL. The assay plates were incubated at room temperature with shaking for 1 h, and the affinity beads were washed with wash buffer (1× PBS, 0.05% Tween 20). The beads were then resuspended in elution buffer (1× PBS, 0.05% Tween 20, 0.5 µM nonbiotinylated affinity ligand) and incubated at room temperature with shaking for 30 min. The kinase concentration in the eluates was measured by qPCR. Results for the primary screen binding interactions were reported as '% Ctrl', where lower numbers indicate stronger hits in the matrix. Selectivity Score or S-score is a quantitative measure of compound selectivity. It is calculated by dividing the number of kinases that compounds bind to by the total number of distinct kinases tested, excluding mutant variants. S = number of hits/number of assays and this value is calculated using % Ctrl as a potency threshold (below) and provides a quantitative method of describing compound selectivity to facilitate comparison of different compounds. S(35) = (number of non-mutant kinases with %Ctrl <35)/(number of non-mutant kinases tested); S(10) = (number of non-mutant kinases with %Ctrl <10)/(number of non-mutant kinases tested), and S(1) = (number of non-mutant kinases with %Ctrl <1)/(number of non-mutant kinases tested).

**Radioactivity metabolite characterization**. MDA-MB-468 tumor xenografts were established on the forelimbs of female athymic nude mice (Hsd:Athymic Nude-Foxn1nu; Envigo; 6 weeks old; RRID:MGI:5652489) by subcutaneous (s.c.) injection of $5 \times 10^6$ cells in a 200 μl cell suspension of a 1:1 v/v mixture of PBS with reconstituted basement membrane (BD Matrigel™, Collaborative Biomedical Products Inc., Bedford, MA). When the tumors reached approximate size of 150 mm³ (~4 weeks), [¹³¹I]-PU-HZ151 (1000 μCi in 150 μL of 5% ethanol in saline, Hospira, 0.9% sodium chloride in water) was injected intravenously into mice. After 24 h, tumors were harvested and weighed prior to homogenization in acetonitrile/H₂O (3:7) using Bullet Blender Tissue Homogenizer (Next Advance Inc.). Tumor-associated radioactivity was extracted in methylene chloride, and the organic layer was separated and dried under vacuum. Samples were reconstituted in 10% CH₃CN in 0.1% HCOOH in water were analyzed using Shimadzu (Columbia, Maryland USA) HPLC system equipped with binary pumps LC-20AB, UV detector and sodium iodide radioactivity detector connected to flow-ram (Lab Logic, Tampa, FL) and 250 × 4.6 mm Phenomenex Luna (Torrance, CA) C-18 HPLC column (10 μM, 100 A). The solvent system consisted of 0.1% Trifluoroacetic acid in water (A), and acetonitrile (B) at a flow rate of 1 mL min⁻¹ and the gradient of 5–60% B from start −23 min A tracer standard ([¹³¹I]-PU-HZ151) co-injected with cold standard of PU-HZ151 was assayed by radioHPLC to confirm the PU-HZ151 elution time. Intact [¹³¹I]-PU-HZ151 elutes with retention time of ~16.1 min. Tumor extract analysis confirmed that only [¹³¹I]-PU-HZ151 contributed to tumor-associated radioactivity.

**Metabolic stability**. PU-HZ151 (1 and 10 μM) was incubated with human, CD-1 mouse, Sprague Dawley rat, and beagle dog liver microsomes (0.5 mg protein/ml) and appropriate cofactors (2.5 mM NADPH and 3.3 mM MgCl₂) in 100 mM phosphate buffer, pH 7.4, in a 37 °C shaking water bath. The incubation contained a final organic solvent concentration of 0.1% DMSO. Reactions were started with the addition of NADPH/magnesium chloride mix and stopped by removing 100 μL aliquots at selected time points (15, 30, and 60 min) and mixing with 200 μl of acetonitrile containing 100 ng/ml haloperidol (internal standard). Following brief vortexing and centrifugation, an aliquot of the supernatant was transferred to a 96-well plate and further diluted 2-fold (for 1 μM, using 10% acetonitrile containing internal standard) or 20-fold (for 10 μM, using 37% acetonitrile containing internal standards) for subsequent LC-MS/MS analysis. Experimental controls consisted of: a) incubation with 1 or 5 μM propranolol as a positive control for metabolism, and b) incubation of 1 and 10 μM PU-HZ151 with heat-inactivated microsomes (0.5 mg protein mL⁻¹) for 0 and 60 min. All samples were assayed in duplicate. To determine metabolic stability of PU-HZ151, the percent remaining at each time point was calculated by dividing the peak area ratio of PU-HZ151 /internal standard at each time point by the mean peak area ratio at 0 min, multiplied by 100.

**Metabolite characterization**. Preparation of samples. In order to detect and characterize metabolites of PU-HZ151 formed in selected microsomal incubation samples, the following procedures were used to prepare the samples: (1) microsomal incubation samples generated in the metabolic stability experiments (1 and 10 μM PU-HZ151, duplicate replicates per timepoint) were initially processed by addition of 2 parts acetonitrile containing the internal standard haloperidol, to 1 part sample, followed by vortexing, centrifugation, and transfer of the supernatant to a 1.5 mL microcentrifuge tube. Samples were stored at −80 °C prior to analysis for metabolites. (2) Samples were thawed to room temperature, vortexed, centrifuged, and a volume of the supernatant diluted 1:1 with 5% acetonitrile, to reduce the percent acetonitrile in the final sample. Samples were then transferred to HPLC vials with glass inserts for subsequent LC-MS analysis.

**Cytochrome P450 enzymes metabolizing PU-HZ151**. To identify cytochrome P450 enzymes responsible for PU-HZ151's metabolism in human liver microsomes the following procedure was followed. Phosphate buffer (100 mM, pH 7.4) was prepared monthly and stored at 4 °C. Frozen stock solutions of human liver microsomes (Life Technologies; Cat# HMMCPL) were used once after thawing and were not refrozen. NADPH stock solution (50 mM) in distilled water was prepared fresh daily. Stock solutions of PU-HZ151 (1 mM in DMSO) and CYP inhibitors, furafylline (10 mM) for CYP1A2, tranylcypromine (2 mM) for CYP2A6, ketoconazole (1 mM) for CYP3A, troleandomycin (20 mM) for CYP3A4, orphenadrine (50 mM) for CYP2B6, montelukast (3 mM) for CYP2C8, sulfaphenazole (10 mM) for CYP2C9, benzylnirvanol (1 mM) for CYP2C19, quinidine (1 mM) for CYP2D6, and 1-aminobenzotriazole (1 mM) for all the P450 enzymes were stored at −20 °C. The stock solutions of these P450 inhibitors were prepared in DMSO. General procedure: PU-HZ151 (1 μM) was incubated with human liver microsomes (0.5 mg protein mL⁻¹), 2.5 mM NADPH, 3.3 mM magnesium chloride, and chemical inhibitors (various concentrations) in phosphate buffer (100 mM, pH 7.4) at 37 °C for up to 60 min. The incubation contained a final organic solvent concentration of 0.1% DMSO. The concentration of the chemical inhibitors used in the assay were furafylline (10 μM), tranylcypromine (2 μM), ketoconazole (1 μM), troleandomycin (20 μM), orphenadrine (50 μM), montelukast (3 μM), sulfaphenazole (10 μM), benzylnirvanol (1 μM), quinidine (1 μM), and 1-aminobenzotriazole (1 mM) for all the P450 enzymes. Timing of the reactions was started following the addition of NADPH/magnesium chloride mix and PU-HZ151. Time-dependent chemical

inhibitors (furafylline, troleandomycin, orphenadrine, and 1-aminobenzotriazole) were preincubated with human microsomes in the presence of NADPH and magnesium chloride for 15 min before the addition of PU-HZ151 and timing of reactions was started after addition of PU-HZ151. The reactions were stopped by removing 100 μL aliquots at selected time points (0, 15, 30, and 60 min) and mixing with 200 μL of cold acetonitrile. Following brief vortexing and centrifugation, a 200 μL aliquot of the supernatant was transferred to a tube, and 10 μL of haloperidol (1 μg mL⁻¹) as internal standard was added. Following brief vortexing and centrifugation, 50 μL of supernatant was added to a 96-well plate for subsequent LC-MS/MS analysis. PU-HZ151 was similarly tested as above in the absence of CYP inhibitors for comparative purposes. All samples were assayed in duplicate. To determine the effect of each CYP inhibitor on the metabolic stability of PU-HZ151, the percent PU-HZ151 remaining at each time point was calculated by dividing the peak area ratio of PU-HZ151/internal standard at each time point by the mean peak area ratio at 0 min, multiplied by 100. This was similarly performed for incubations of PU-HZ151 performed in the absence of CYP inhibitor.

**LC-MS methods for metabolite characterization**. Samples were analyzed by LC-MS using positive-ion electrospray ionization and reverse-phase chromatography in full-scan mode with MS/MS product-ion spectra generated for detected metabolite masses. This was performed using the MS2 scan mode of the G6410B mass spectrometer to produce MS/MS product-ion spectra for masses detected at sufficient intensities. Chromatography was performed using a ZORBAX Eclipse XDB-C18 4.6 × 50 mm, 5 μm, column, with gradient elution at 1 mL min⁻¹. Mobile phase was comprised of A = MilliQ Water with 0.1% formic acid, B = acetonitrile. Six metabolites were detected (M1, M2, M3a, M3b, M4, M5) and chromatography resolved.

**PAMPA-BBB**. Each well of a 96-well Teflon tray acceptor plate (Millipore, MA, USA) was filled with 300 μL universal PBS buffer (pH 7.4)[38]. The 96-well PVDF donor plate (0.45 μm pore size) was applied 4 μL of porcine polar brain lipid (PBL) in dodecane (20 mg mL⁻¹). Immediately after the lipid wetted the membrane, 250 μL of test compound solution (50 μM) in triplicate was added to the wells of the donor plate. The donor and acceptor plates were assembled into a sandwich and incubated at room temperature for 16 h. The donor plate was carefully removed and 200 μL of the solution in the acceptor plate was placed on a 96-well UV-transparent Microplate. The absorbance was measured by UV spectroscopy on the GT Analyst (Molecular Devices) plate reader at 260 nm. A reference solution was prepared at the same concentration as the sample solution without membrane. The permeability coefficient was calculated using the following equation: $P_e = C \times (-) \ln(1 - [\text{drug}]_{\text{acceptor}}/[\text{drug}]_{\text{reference}})$; $C = V d \times V a/(V d + V a) \times \text{area} \times \text{time}$ (where Va: volume of acceptor plate; Vd: volume of donor plate; area: membrane area (0.266 cm²); time: incubation time). Desipramine was used as a CNS-high permeability standard ($P_e = /12 \times 10^{-6}$ cm s⁻¹, reported value; $16.4 \pm 2.2 \times 10^{-6}$ cm s⁻¹ determined in our hands) and caffeine was used as low CNS-permeability standard ($P_e = /1.3 \text{ x}/10^{-6}$ cm s⁻¹ reported value; $4.7 \pm 0.1 \times 10^{-6}$ cm s⁻¹ determined in our hands). Each drug was tested in triplicate.

**In silico prediction of molecular properties**. Calculated physicochemical properties were obtained using standard commercial and freeware packages: Schrödinger software suite release 2018-1 (QikProp and Epik), ChemAxon Marvin suite (MarvinSketch 20.19) and VCCLAB (ALOGPS 2.1). The 2D structures of all the title compounds were generated by using ChemBioDraw Ultra 12.0. LigPrep module implemented in Schrödinger was used to generate energy minimized 3D structures. Partial atomic charges were computed using the OPLS_2005 force field and all compounds were prepared in neutralized form for the calculation of molecular properties[58]. The molecular weight (mol_MW), polar surface area (PSA), hydrogen bond donor (HBD) and octanol/water partitioning coefficient (QPlogP_o/w) were calculated using QikProp in Schrödinger's Maestro 11.5[59]. For the most basic $pK_a$ calculations (Cp$K_a$), the 3D structures (neutral form) were then subjected to $pK_a$ prediction at pH 7.4 (solvent = water) using the sequential mode of the Epik program version 4.3011[60,61]. The water-octanol distribution coefficient at pH 7.4 (Clog$D_{7.4}$) was calculated using the ChemAxon method of MarvinSketch 20.19 (Budapest, Hungary)[62–64]. The Web site of VCCLAB provides free online chemoinformatics tools for the calculation of various physiochemical properties. The ALOGPS 2.1 software was used for the online prediction of log P (AlogP) and water solubility (logS)[65–67]. The Central Nervous System Multiparameter Optimization (CNS MPO) scores were calculated using the reported template[34,68].

**Organotypic explant culture of GBM**. GBM tumor specimens were obtained from the surgical suite under an approved protocol (IRB # 09-156, PI: Viviane Tabar, MD). GBM explants were prepared as described previously[47,69]. Briefly, GBM tumor specimens were collected from the surgical suite and placed on ice. The tumor tissue was treated with red blood cell (RBC) lysis buffer (eBioscience, San Diego, CA) and washed with ice-cold sterile phosphate buffered saline (PBS, pH 7.4) containing 3X penicillin (300 IU mL⁻¹) and 3X streptomycin (300 μg mL⁻¹). After removing necrotic tissue, the tumor tissue was carefully cut into small explants (~2 mm³) using a pair of sterile scalpels. Explants were placed atop Millicell inserts (up to six explants per insert) (Millipore, Billerica, MA) pretreated

with 1 µg mL$^{-1}$ fibronectin (BD Biosciences, Franklin Lakes, NJ), and cultured in 35 mm dishes containing 1 mL N2 medium consisting of F12-DMEM with N2 supplements (Life Technologies, Carlsbad, CA), 1.55 mg mL$^{-1}$ glucose and 2 mg mL$^{-1}$ sodium bicarbonate. The medium was changed every other day. Agents used to treat the explants include: PU-H71, PU-HZ151, temozolomide (TMZ) (Sigma Aldrich, St. Louis, MO) and bevacizumab (Genentech/Roche, Mahwah, NJ). Radiation (10 Gy) was delivered via X-RAD 225 C irradiator (Precision X-Ray Inc., North Branford, CT). The explants were harvested at 10 days after start of treatment with the drugs. For radiation, the explants were treated on day 5, and harvested on day 10.

**Western blotting**. Protein extracts were prepared in 50 mM Tris-HCl, pH 7.4, 150 mM NaCl, 1% NP40 buffer supplemented with protease inhibitors cocktail (Roche). The samples were resolved by SDS-PAGE, transferred onto nitrocellulose membrane and probed with primary-, followed by horseradish peroxidase-conjugated secondary antibody. Primary antibodies used: HSP90β (H90-10) (SMC-107; RRID:AB_854214; 1:3000) and HSP110 (SPC-195; RRID:AB_2119373; 1:2000) were purchased from StressMarq; HSP90α (ab2928; RRID:AB_303423; 1:6000) and Olig2 (ab109186; RRID:AB_10861310; 1:1000) from Abcam; β-actin (A1978; RRID:AB_476692; 1:3000) from Sigma-Aldrich; p53 (2524; RRID: AB_331743; 1:2000), PTEN (9188; RRID:AB_2253290; 1:1000), Bcl-2 (15071; RRID:AB_2744528; 1:1000), EGFR (2085; RRID:AB_1903953; 1:1000), PDGFRα (3174; RRID:AB_2162345; 1:1000), p-AKT (Ser473) (4060; RRID:AB_2315049; 1:1000), p-AKT (Thr308) (13038; RRID:AB_2629447; 1:1000), AKT (4691; RRID: AB_915783; 1:2000), p-ERK1/2 (Thr202/Tyr204) (4370; RRID:AB_2315112; 1:2000), ERK 1/2 (9102; RRID:AB_330744; 1:4000), Bim (2933; RRID: AB_1030947; 1:1000), Bad (9268; RRID:AB_10695002; 1:1000), Bid (2002; RRID: AB_10692485; 1:1000), Bcl-xL (2764; RRID:AB_2228008; 1:1000), Bcl-w (2724; RRID:AB_10691557; 1:1000), Mcl-1 (5453; RRID:AB_10694494; 1:1000), GAPDH (5174; RRID:AB_10622025; 1:5000), RB (9309; RRID:AB_823629; 1:1000), p-RB (Ser807/811) (8516; RRID:AB_11178658; 1:1000), NF1 (14623; RRID:AB_2798543; 1:1000), HOP (5670; RRID:AB_10828378; 1:2000), EGFR (4267; RRID: AB_2246311; 1:1000), Pgp, also called multi-drug resistance protein 1 (MDR1) (12683; RRID:AB_2715689; 1:1000) and CDC37 (4793; RRID:AB_10695539; 1:2,000) from Cell Signaling Technology; HSP70 (ADI-SPA-810; RRID: AB_10616513; 1:2,000), HOP (ADI-SRA-1,500; RRID:AB_10618972; 1:2000) and HSC70 (ADI-SPA-815; RRID:AB_10617277; 1:4,000) from Enzo and cleaved PARP, and p85 fragment (G7341; RRID:AB_430876; 1:1000) from PromegaBlots were visualized by enhanced chemiluminescence (GE). ImageJ (versions 1.4 and 1.52) was used for western blot quantification.

**Glioma stem-like cell culture and neurosphere assay**. Single cell suspension of GBM tumor cells was prepared as described previously[46]. Briefly, tumor tissue was treated with RBC lysis buffer and washed with ice-cold sterile PBS containing 3X antibiotics. The tissue was then minced into small pieces (<1 mm$^3$) with a pair of sterile scalpels, and then incubated in Hank's balanced salt solution without Ca$^{2+}$ and Mg$^{2+}$ (CMF-HBSS) for 5 min at 37 °C, and then in 1X Accumax (Innovative Cell Technologies Inc., San Diego, CA) for 10 min at room temperature. The tissue was resuspended in ice-cold CMF-HBSS, and the tumor cells were released from the tissue by gentle pipetting. The cells were cultured in low-attachment culture plates (Corning, Corning, NY) with EGF and FGF (20 ng mL$^{-1}$ each) and B27 supplement (1:50; Gibco, Carlsbad, CA) in N2 medium to form NSs. To establish an adherent GSC culture[70], NSs were plated on culture dishes treated with 15 µg mL$^{-1}$ polyornithine (PO), 4 µg mL$^{-1}$ mouse laminin 1, and 2 µg mL$^{-1}$ fibronectin, and cultured in N2 medium containing EGF and FGF. For serial NS assays, primary NSs were dissociated with Accumax, and plated at 500–1000 cells per well in a low-attachment six-well plate in N2 medium containing EGF, FGF, B27 supplement, and 0.15 % methylcellulose to minimize cell aggregation.

**Immunohistochemistry and TUNEL assay**. GBM explants were fixed in ice cold 4% paraformaldehyde (pH 7.4) overnight at 4 °C, washed twice with PBS, then transferred to 30% sucrose in PBS and incubated at 4 °C overnight. The explants were embedded in Tissue-Tek Optimal Cutting Temperature (OCT) compound (Sakura Finetek, Torrance, CA) and sectioned at 10 µm on a cryostat (Leica, Deerfield, IL). After washing in PBS, sections were blocked in PBS containing 10% normal goat serum, 1% bovine serum albumin, and 0.1% Triton X-100 for one hour. The sections were then incubated with primary antibodies at 4 °C overnight, washed three times with PBS, incubated with the appropriate secondary antibodies (1:400, Alexa conjugates, Molecular Probes, Invitrogen) for two hours at room temperature, counterstained with DAPI (4′,6-diamidino-2-phenylindole, Molecular Probes) and mounted in 70% glycerol. Primary antibodies included: Ki67 (1:100 Dako) (GA2661-2; RRID:AB_2687921), CD105 (1:100, Dako) (M3527; RRID: AB_2099044), CD31 (1:100, BD Biosciences) (550274; RRID:AB_393571), cleaved Caspase-3 (9661; RRID:AB_2070042) (1:100, Cell Signaling Technology). Apoptotic cells were detected by a subsequent TUNEL (Terminal deoxynucleotidyl transferase dUTP nick end labeling) using In Situ Cell Death Detection Kit, Fluorescein (Roche Applied Sciences, Indianapolis, IN), after the DAPI staining step.

**Viability assay**. GSCs were adherently cultured in PO/laminin/fibronectin-coated 96 or 48-well plates, and treated with various concentrations of inhibitors in N2 medium containing EGF and FGF in duplicates. At 0 h and 48 h post treatment, cells were harvested in CellTiter-Glo lysis buffer, and viability (based on ATP level) was measured according to the manufacturer's instruction for CellTiter-Glo cell viability assay (Promega, Madison, WI). Percentage of growth inhibition was calculated by comparing luminescence values obtained from treated versus control cells, accounting for initial cell population at time zero. To measure GBM explant tissue viability, each explant was homogenized in a buffer consisting of PBS/CellTiter-Glo lysis buffer (1:1) and was mixed with an equal volume of CellTiter-Glo reagent to measure viability. Viability from at least three explants was measured for each treatment group. MDA-MB-468 and HepG2 cells were plated in 96-well black plates at 8000 cells per well. The compounds were added to the cells in triplicates by serial dilutions and incubated for 72 h. Cell viability was measured by CellTiter-Glo luminescent cell viability assay (Promega, Madison, WI) and analyzed as described above.

**Tumor-derived endothelial cell culture**. Single cell suspension was prepared from a freshly resected GBM tumor as described above, and CD105 + tumor endothelial cells were isolated by a MoFlo cell sorter (Beckman Coulter, Pasadena, CA) using a FITC-conjugated CD105 antibody (1:20, BD Biosciences) (561443; RRID: AB_10714629). The acquired data was saved into a flow cytometry standard file and analyzed using Flow Jo v10.7 software (FlowJo LLC.). Cells were resuspended in endothelial growth medium consisting of endothelial basal medium-2 (EBM-2) (Lonza, Allendale, NJ), 2.5% FBS, 10 ng mL$^{-1}$ FGF2, and 100 µg mL$^{-1}$ gentanamicin, and plated on 96-well plates pretreated with rat tail collagen type I. After 4 days, the medium was switched to an endothelial differentiation medium consisting of EBM-2, 0.25 % FBS, 100 nM hydrocortisone, and 100 µg mL$^{-1}$ gentanamicin, and cultured for another 4 days. The cells were then treated with various concentrations of inhibitor in the endothelial differentiation medium for 6 days before they were harvested for viability assay, as described above.

**Tumor establishment**. For subcutaneous xenografts, mice were anesthetized with 2% isoflurane (Baxter Healthcare) (2 l min$^{-1}$ medical air). MDA-MB-469 and ASPC-1 cells were implanted subcutaneously (5 × 10$^6$ cells in 150 µL 1:1 PBS/matrigel® (BD Biosciences, San Jose, CA) in the right shoulder or flank and allowed to grow for ~2 weeks until the tumors reached 5–10 mm in diameter. For orthotopic xenografts, mice were anesthetized with 2% isoflurane (2 L min$^{-1}$ medical air) (Baxter Healthcare), and Boreholes were made in the skull using a small hand drill. Patient-derived glioma-stem cells and MDA-MB-468 cells (5 × 10$^5$ cells in 2 µL PBS) were stereotactically injected into the striatum (2 mm lateral and 1 mm anterior to the bregma; 2.7 mm deep) using a Stoelting Digital New Standard Stereotaxic Device and a 5 µL Hamilton syringe. Cells were allowed to grow for 3–6 weeks.

**Small-Animal PET imaging**. Orthotopic GBM xenografts in mice were established by injection of 5 × 10$^5$ cells in 2 µL PBS. For PET analyses, GSCs were injected into the striatum. Boreholes were made using a small hand drill. Mice were imaged with MRI (typically ~ 2–3 weeks after injection), and – if tumors were visible on MRI – they were imaged by micro-PET using the [124I]-radiolabeled PU-HZ151. When mice started showing symptoms of progressing GBM, they received a microdose of [124I]-PU-HZ151 and serial PET scans were performed at 20 mins, 6, 24, 48, 72, and 96 h post tracer administration using a dedicated small-animal PET scanner (Focus 120 microPET; Siemens Medical Solutions, USA, formerly Concorde Microsystems, Knoxville, TN). Mice were maintained under 2% isoflurane (Baxter Healthcare, Deerfield, IL) anesthesia in oxygen at 2 L min$^{-1}$ during the entire scanning period. To reduce the thyroid uptake of free iodide arising from metabolism of tracer, mice received 0.01% potassium iodide solution in their drinking water starting 48 h prior to tracer administration. For PET imaging, each mouse was administered [124I]-PU-HZ151 (15 µCi g$^{-1}$, 3 µCi µL$^{-1}$ in in saline, Hospira, 0.9% sodium chloride in water) via the tail vein or ip. Approximately 5 min prior to PET acquisition, mice were anesthetized by inhalation of a mixture of isoflurane (Baxter Healthcare, Deerfield, IL, USA; 2% isoflurane, 2 L min$^{-1}$ medical air) and positioned on the scanner bed. Anesthesia was maintained using a 1% isoflurane-O$_2$ mixture. PET data for each mouse was recorded using static scans of 15 min and acquired at the indicated times post injection. Quantification and %ID g$^{-1}$ values were calculated by manually drawing regions of interests in three different frames and determining the average values using ASI Pro VM™ MicroPET Analysis software (Siemens Medical Solutions, Knoxville, TX).

**In vivo biodistribution**. For acute in vivo biodistribution studies, mice with xenografts on forelimbs were injected intravenously in the tail vein with 25–50 µCi of [131I]-PU-HZ151 in 200 µL of saline, Hospira, 0.9% sodium chloride in water. Activity in the syringe before and after administration was assayed in a dose calibrator (CRC-15R; Capintec) to determine by difference the actual activity administered to each animal. Animals were euthanized by CO$_2$ asphyxiation at different time points post-administration of the tracer and organs including tumors were harvested and weighed. I-131 was measured in a scintillation g-counter (Perkin Elmer 1480 Wizard 3 Auto Gamma counter, Waltham, MA) using a 260–340 keV energy window. Count data were background and decay-corrected to

the time of injection, and the percent injected dose per gram (%ID g$^{-1}$) for each tissue sample was calculated by using a measured calibration factor to convert count rate to activity and the activity was normalized to the activity injected and the mass of sample assayed to yield the activity concentration in %ID g$^{-1}$.

**Pharmacokinetic studies in mice.** Pharmacokinetic studies were performed in male B6D2F1 mice (Harlan Laboratories) following oral (po) or intravenous (iv) administration of PU-HZ151. Blood and brain tissue were collected following euthanasia at the scheduled time points of 5, 30 min, 1, 4, and 8 h following iv dosing and of 30 min, 1, 4, and 8 h following po dosing. Brain samples were immediately flash frozen in liquid nitrogen following collection and stored at −80 °C until analysis. Blood samples were collected into an EDTA tube by heart puncture and centrifuged at 17,000×g for 5 min at 4 °C to separate the plasma and immediately frozen on dry ice and stored at −80 °C until analysis. The plasma sample (50 μL) was added with 250 μL of acetonitrile and 10 μL of internal standard (PU-HZ151-$d_4$), and then thoroughly vortexed to precipitate plasma protein and centrifuged at 17,000×g for 5 min at 4 °C. The supernatant was evaporated on a GeneVac and the residue was taken up into 100 μL of mobile phase (65% H$_2$O: 35% acetonitrile + 0.1% FA), applied to a 96-well plate and analyzed by LC-MS/MS (6410, Agilent Technologies) to measure concentrations of PU-HZ151. A Zorbax Eclipse XDB-C18 column (4.6 × 50 mm, 5 μm) was used for the LC separation, and the analyte was eluted under an isocratic condition (65% H$_2$O + 0.1% HCOOH: 35% CH$_3$CN) for 5 min at a flow rate of 0.4 mL/min. Concentration in plasma was determined by comparison to a standard curve prepared by spiking untreated plasma samples to appropriate concentration of PU-HZ151 (1–2000 ng mL$^{-1}$), processed and analyzed in the same manner as treated samples specified above. Frozen brain tissue was dried, weighed, and added with 750 μL of water/acetonitrile (7:3) and 10 μL of internal standard (PU-HZ151-d$_4$) solution. This was homogenized, extracted with methylene chloride, and evaporated on a GeneVac. The residue was taken up into 100 μL of mobile phase (65% H$_2$O: 35% acetonitrile + 0.1% FA) and analyzed by LC-MS/MS (6410, Agilent Technologies) to measure concentrations of PU-HZ151. A Zorbax Eclipse XDB-C18 column (4.6 × 50 mm, 5 μm) was used for the LC separation, and the analyte was eluted under an isocratic condition (65% H$_2$O + 0.1% HCOOH: 35% CH$_3$CN) for 5 min at a flow rate of 0.4 mL min$^{-1}$. Concentrations in tissues were determined by comparison to a standard curve of PU-HZ151 (1–2000 ng mL$^{-1}$). Drug concentration (μM) in the brain was then modified to account for an average water space of 0.8 mL g$^{-1}$.

**Epichaperome detection in human patients.** [$^{124}$I]-PU-HZ151 and [$^{124}$I]-PU-H71 were synthesized in-house by the institutional cyclotron core facility at high specific activity. Analyses of the epichaperome by positron emission tomography (PET-CT) were performed as previously reported[13]. In brief, research PET-CT was performed using an integrated PET-CT scanner (Discovery DSTE, GE Healthcare). CT scans for attenuation correction and anatomic coregistration were performed before tracer injection. Patients received 5 mCi of [$^{124}$I]-PU-HZ151 or [$^{124}$I]-PU-H71 by peripheral vein over 2 min. PET data were reconstructed using a standard ordered subset expected maximization iterative algorithm. Emission data were corrected for scatter, attenuation, and decay. PET scans were performed 0.5, 3, and 24 h after tracer administration. Numbers in the scale bar indicate upper and lower standardized uptake value (SUV) thresholds that define pixel intensity on PET images. SUV is a pixel value, derived from PET measurements of in vivo radioactivity, quantified as the fraction of the radioactive dose injected found in a cubic centimeter of measured tissue multiplied (ie, normalized) by patient body weight (grams). Maximum Intensity Projection (MIP) 3D images were constructed from the voxels with maximum intensity that fell in the way of parallel rays traced from the viewpoint to the plane of projection. At each pixel the highest data value encountered along the corresponding viewing ray was determined. MRI scans were obtained as part of routine clinical care using hardware and standard data acquisition and processing methods. MRI was obtained using a 1.5T magnet (Signa HDxt, GE Healthcare, Milwaukee, WI). Images were acquired using 5 mm slice thickness with no interslice gap. axial fast spin-echo T2-weighted (repetition time [TR]/echo time [TE] = 4000/100 msec, matrix 256 × 256); axial fluid-attenuated inversion recovery (FLAIR; TR/TE/inversion time [TI] = 10,000/160/2200 msec, matrix 256 × 256); sagittal and axial T1-weighted; and contrast coronal, sagittal, and axial T1-weighted images (TR/TE = 500/10 msec, matrix 256 × 256) were obtained. Gadopentetate dimeglumine (Magnevist, HealthCare Pharmaceuticals Inc.) was injected though a peripheral angiocatheter at a standard dose (0.2 mL kg$^{-1}$ body weight, 40 mL, 2 cc s$^{-1}$). For precise localization of tracer uptake to specific neuroanatomical structures, PET brain images were fused to MRI brain images using the Integrated Registration application of the AW Suite software package (GE Healthcare, Milwaukee, Wisconsin, USA).

**Plasma pharmacokinetics assays in human patients.** All patient blood samples had a valid chain of custody from venipuncture through onsite specimen analysis and disposal. For [$^{124}$I]-PU-HZ151 injection and blood sampling, separate peripheral intravenous lines were placed. Blood samples were collected 1, 5, 15, 30, 60–90, and 150–270 min post injection. For quantification of radioprobe clearance and metabolism, plasma samples were radioassayed by well counter and radioHPLC using standard techniques. Activity in whole blood and plasma

specimens was radioassayed by well counter (1480 WIZARD® 3″ Automatic Gamma Counter, PerkinElmer, Shelton, Connecticut USA), after separating blood and plasma by centrifuge (3000×g for 10 min at 4 °C). Well counting obtained a minimum of 10,000 counts per specimen correcting for background activity and radioactive decay from time of tracer injection. An I-124 standard was assayed on the well counter to obtain a calibration factor for converting sample scintillation count rates (in cpm) to millicurie (mCi) of activity. Dividing sample activity by the sample volume and intravenously administered patient dose (mCi) quantified the sample activity in terms of fraction-of-injected-dose (expressed as a decimal) per mL (ID mL$^{-1}$); multiplying this quantity by 100% yields the percent of the injected dose per ml (%ID mL$^{-1}$). This value was multiplied by the patient body weight (g) to convert to standard uptake values (SUV) for comparison with tissue SUV values measured by PET, also calculated as the ID per mL tissue volume, normalized to patient body weight (g). Blood and plasma concentration data were separately plotted versus time. A bi-exponential mathematical function was fit to the experimental data by a least-squares approach, with goodness-of-fit assessed by conventional graphic and coefficient analyses. From the patient-specific fitted curve formulae, individual blood and plasma clearance half-times and other pharmacokinetic parameters were determined. The software used for the fits and to derive the PK parameters was EXCEL 365 (Microsoft Corp, Redmon, WA).

**Digital autoradiography and H&E staining.** For epichaperome detection via autoradiography, tumor bearing mice were administered with [$^{131}$I]-PU-HZ151 (15 μCi g$^{-1}$, 3 μCi μL$^{-1}$ in saline, Hospira, 0.9% sodium chloride in water) via tail vein injection. Prior to brain excision, mice were perfused with 20 mL PBS. Brains were excised at the indicated time post injection and fixed in Tissue-Tek O.C.T. compound (Sakura Finetek, Torrance, CA) and flash frozen in liquid nitrogen and cut into 10 μm sections using a Vibratome UltraPro 5000 Cryostat (Vibratome, St. Louis, MO). A storage phosphor autoradiography plate (Fujifilm, BAS-MS2325, Fiji Photo Film, Japan) was exposed to the tissue slices overnight at −20 °C and read the following day. Relative count intensity of the sections in each image was quantified using ImageJ 1.47u processing software. The same sections were subsequently subjected to hematoxylin-eosin (H&E) staining for morphological evaluation of tissue pathology and to compare the localization of the epichaperome probe with the location of tumor tissue.

**Epichaperome detection by native gel.** Cultured cells and orthotopic tumors were collected and lysed in 20 mM Tris pH 7.4, 20 mM KCl, 5 mM MgCl$_2$, 0.01% NP40 buffer containing protease, and phosphatase inhibitors. Protein extracts were loaded onto 4–10% native gradient gel and resolved at 4 °C. Samples were transferred onto nitrocellulose membrane in 0.1% SDS-containing transfer buffer for 1 h. The membranes were immunoblotted with the following antibodies: HOP (SRA-1500, Enzo; RRID:AB_10618972; 1:2000); HSP90α (ab2928, Abcam; RRID: AB_303423; 1:6000); HSP90β (SMC-107, StressMarq; RRID:AB_854214; 1:3000) and β-actin (A5316, Sigma; RRID:AB_476743; 1:3000).

**Survival studies in mice.** Following intracranial xenotransplantation of the GSCs, and 7 days after tumor transplantation (5 × 10$^5$ cells in 2 μL PBS) mice were treated with PU-HZ151 (50 mg kg$^{-1}$ of PU-HZ151 in 5 mM citrate buffer + 15% captisol, iv on a Monday-Wednesday-Friday schedule) or vehicle three times per week. Survival of the animals was followed until the mice became morbid (physical and/or neurological symptoms), at which point, the animals were killed. Typically, mice became symptomatic several weeks to a few months after GSC implantation. Log-rank tests were used to compare the distribution of time to killing across groups and Kaplan–Meier curves were produced.

**Statistics and reproducibility.** Statistics were performed and graphs generated using Prism 8 or 9 software (GraphPad). Statistical significance was determined using either Student's t-tests (2-group comparisons) or ANOVA (multiple comparisons). The survival benefit was determined using a log-rank (Mantel–Cox) test. Pearson's coefficient was determined to measure the statistical relationship between variables. Means and standard errors were reported for all results unless otherwise specified. Effects achieving 95% probability (i.e., $p < 0.05$) were interpreted as statistically significant. No statistical methods were used to pre-determine sample sizes, but these are similar to those generally employed in the field. Unless otherwise indicated, experiments were repeated three times with similar results.

**Reporting summary.** Further information on research design is available in the Nature Research Reporting Summary linked to this article.

## Data availability

The source data underlying all main and supplementary figures are provided as a Source Data file. All data generated or analyzed during this study are included in this published article (and its supplementary information files). For the pilot feasibility clinical study, the supplementary information contains the Study protocol. De-identified raw data from individual patients enrolled on the pilot feasibility clinical study can be obtained from M. P.D. (dunphym@mskcc.org) upon reasonable request. These data include time-dependent biodistribution of [$^{124}$I]-PU-AD in blood, major organs, tissues-of–interest,

and excreta, including blood-based characterization of tracer metabolism. Source data are provided with this paper.

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

## Acknowledgements

We thank Michael Metzner for his help with establishing tumor tissue cultures and Raj Mathews for his help with immunohistochemistry. This work was supported by the U01 AG032969, R56 AG061869, R56 AG072599, R01 AG067598, R01 CA172546, R01 AG067598, Coins for Alzheimer's Research, ADDF, the Mr. William H. Goodwin and Mrs. Alice Goodwin and the Commonwealth Foundation for Cancer Research and the Experimental Therapeutics Center of the Memorial Sloan Kettering Cancer Center, the MSKCC Brain Tumor Center, P30 CA008748 (NCI Core Facility Grant) and U54 OD020355. Technical services provided by the MSKCC Small-Animal Imaging Core Facility, supported in part by NIH Small-Animal Imaging Research Program Grant No R24 CA83084 are gratefully acknowledged. The sponsor of the pilot clinical study, Samus Therapeutics, Inc. (Samus), has reviewed this manuscript and agrees with the findings of this study, supporting the public disclosure of the study results. Samus was not involved with the design of the study protocol, the collection of study data, the analysis of study data or the writing of this manuscript.

## Author contributions

A.B. and K.P. designed and performed the imaging probe discovery and synthesis, the epichaperome detection by PET and autoradiography, and the survival studies. D.Z., H.H., H.J.P., S.O.O., M.R.P., L.S., W.S., B.S.I., S.S., C.S.D., R.K., H.T., and T.T. designed and performed the medicinal chemistry studies. F.S. performed the GBM studies. M.P.D. and A.Rimner were investigators of the PET imaging clinical trial study. P.Z., M.G., and B.J.B. performed clinical data analyses. S.L. and E.B. performed radiotracer synthesis for the human PET imaging studies. S.J., S.K.S., S.W., A.Rodina, P.D.P., A.R.M., P.Y., P.P., J.A., A.C., A.K., T.K., and S.G.L. performed experiments. T.T., M.G., J.S.L., S.L., and V.T. participated in the design and analysis of various experiments, and G.C., T.T., S.D.G., and F.S. wrote the paper.

## Competing interests

Memorial Sloan Kettering Cancer Center holds the intellectual rights to this portfolio (PCT/US2007/072671, Treatment of neurodegenerative diseases through inhibition of HSP90 and WO2013/009655, Uses of labeled HSP90 inhibitors). Samus Therapeutics Inc, of which G.C. has partial ownership, and is a member of its board of directors, has licensed this portfolio. G.C., T.T., H.H., D.Z., A.R., N.P., S.L., and J.L. are inventors on the licensed intellectual property. All other authors declare no competing interests.
