## [Peer Review File · Nature Communications]

REVIEWER COMMENTS

Reviewer #1 (Remarks to the Author):

The paper by Bolaender et al describes the discovery and validation of new chemical tools that target disease-related (glioblastoma in particular) complexes of proteins. The latter form the so called epichaperomes, complex protein-protein assemblies that form around hsp90 and 70.

The paper is very well written and presents an incredible wealth of data and results. There is a great combination of medicinal chemistry, BBB experiments, in vitro data, selectivity tests, in vivo/ex vivo experiments, imaging, which make this work really thorough and complete. The data also show that the new molecules are selective for GBM and may be useful for mechanistic studies.

The only modifications may be suggested in the discussion, where the authors may want to consider discussing the possibility to use molecules that target PPIs in chaperone complexes that have been developed using different concepts than PU-H71, such as for example the ones in [10.1002/chem.201904523](https://doi.org/10.1002/chem.201904523), [10.1002/chem.202000615](https://doi.org/10.1002/chem.202000615).

Overall, I appreciated reading this paper, its in-depth investigation from molecular design to in vivo. I think it is worth publication in Nat. Comms.

Reviewer #2 (Remarks to the Author):

In the manuscript, the authors used chemical tools to overcome the limitation of already known epichaperomes' inhibitor PU-H71. It cannot permeate through blood barrier brain. They found that PU-HZ151 could be a potent chemical probe to pass through BBB and showed the selectivity to epichaperomes. They used isotopes of iodine in the compound structure to do to PET & radiology. For the application, the authors used glioblastoma. Depending on the samples, the formation of epichaperomes were different, but the results showed that PU-HZ151 was selective to the epichaperomes and cell viability. The application and the logic flow of the manuscript are excellent. Therefore, it is a good paper to publish to Nat. Commun. after revising some minor points as below.

[Minor points]

1. I wondered why the authors selected PU-BIS-11 as the final candidate for the replacement of PU-H71, even though it had lower affinity to Hsp90.
2. The authors made xenograft mice using MDA-MB-468 and ASPC1. Then they injected only PU-HZ151, but it will be good to compare PU-H71 together to show the better performance of the selected compound. Also, they did not mention that they used muscle and heart from which xenograft mice.
3. In figure 4C&D, the authors did the autoradiography and H&E staining. The signal of radiography in H&E staining position was weak. Could it be improved?
4. In figure 5C&D, the information is missing which PU compound was used.

Reviewer #3 (Remarks to the Author):

The HSP90 inhibitor, or now “epichaperome complex”, has been proposed for use in cancer and AD for some time. This work offers new methods to screen for compounds that hit targets within the “epichaperome”. This is an interesting framework for imaging a compound that selects for tumors based on structural differences in the arrangement of common chaperone proteins that cause the agent to reside longer in the malignant regions. Claims for applications in the CNS and AD are less convincing than the work towards exploring the hypotheses in oncology. There are a few concerns that need to be addressed prior to publication:

Major concerns:

1. The authors claim that they have developed probes to modulate the epichaperome is too broad and needs to be scaled back from the title, abstract and throughout the manuscript, as the molecules are still designed to bind to one specific target (HSP90). Specifically, the lead compound “151”, appears to have some use in oncology but the applications in AD are not clear. Although the probe is primarily selected for brain penetration, most of the work was carried out in tumors (including the GBM model which was peripheral). It is not justified based on the current experiments to claim that this is a “CNS-directed epichaperome probe”.
 - a. There is a lack of evidence that this approach will be clinical useful in the central nervous system as claimed in the title and throughout the manuscript. For example, what is the ligand binding to in AD? Authors need to also address toxicity issues and lack of in vivo efficacy from this approach.
 - b. Standard neuroimaging evaluations for a new radiopharmaceutical were not considered and need to be included to make claims about CNS and AD. For example:

- i. The uptake in tumors and low uptake in brain is demonstrated. However, specificity is questionable in the CNS in vivo. Why were blocking experiments not performed, in the in vitro autoradiography or in vivo imaging? These are required and should be included.
 - ii. The authors need to show selectivity over other CNS targets in vitro (not just potency to the primary target) for neuroimaging in AD.
 - iii. Brain penetration is low (ca. 1 SUV) and could have been improved. If BBB penetration is a major focus of the medicinal chemistry, the authors would benefit from CNS MPO analyses and should be reported. The authors focused on solubility and lipophilicity (but only logP or logD are discussed and should be discussed in the context of neuro-PET tracers that have log D in the 1.5-3.5 range. The authors have minimal data reported for solubility; "acid/base character" is normally called apparent pKa, which is used later in the same paragraph; there is no mention of efflux transporter such as Pgp in this section either and such assays should also be reported to give insights into the medicinal chemistry design for crossing the BBB.
 - iv. Radiometabolite analysis needs to be done and reported.
- c. In Figure 3b, the Native-PAGE gel (right panel) shows that bands appear for both MB-468 and ASPC-1 samples, with MB-468 cells having a much higher signal compared to ASPC-1, however, without a loading control or any indication of how much total protein was loaded the reader cannot be certain that the MB-468 sample isn't overloaded. The authors state that MB-468 has high expression of epichaperomes while ASPC-1 has low expression, however, there is no loading control probe, such as β -actin to demonstrate that the same amount of sample was loaded into each sample well, as seen in the SDS/PAGE gel. The authors should indicate the amount of total protein loaded into each sample lane to clarify whether the same total protein quantities were used, and should have a loading control to indicate these quantities. The same concern is noted in the Native-PAGE blot in Figure 6c.

Minor concerns:

1. The chemistry was done on a small scale which is acceptable for screening purposes, however, the characterization is incomplete for the standard of Nature or Medicinal Chemistry journals. Specifically, several compounds are missing ^{13}C NMR spectra, HRMS, etc.
2. The nomenclature consensus guidelines should be used for this manuscript (e.g. consistent use of square brackets when needed, mCi vs. Bq, etc).
3. The authors describe the compounds as "probes" even when unlabeled. Are the cold compounds to be used as a tool/probe just to study the epichaperome or therapeutic (if so the authors should justify why a compound that binds to the epichaperome proteins would be expected to be therapeutic). The use of this probe/therapeutic agent may be limited to understanding the role of the epichaperome in tumour vitality. A probe directed at tumour stem cells would still be necessary to determine the efficacy of the treatment.

4. 3) It is not clear whether the dose required for therapy would be detrimental to the "healthy" form of the chaperone proteins. The concentration sufficient to cause apoptosis in the explants is only specified in Fig 5 caption but not the Methods.

5. The abstract is very vague and could benefit from being rewritten. The title should be revised to eliminate the implication that these compounds are designed for the CNS and should also be more specific to the experiments conducted herein.

6. The preclinical MRI scanner make and model, and the weighting of the MRI images (presume T2-weighted) should be added to the experimental details. The sentence "Magnetic resonance imaging (MRI), by offering structural and soft tissue resolution, confirmed the presence of PU-HZ151 in the brain tissue" is misleading. MRI is only providing anatomical detail to co-localize with the PET signal, but this sentence, depending on which word is stressed, implies that MRI is sensitive to the compound.

We thank the Reviewers for their thoughtful and constructive critiques, and are pleased they concluded this manuscript merits publication in *Nature Communications* pending revision. We are also grateful the Reviewers appreciate the large amount of work that went into this manuscript as it “Presents an incredible wealth of data and results...from medicinal chemistry, BBB experiments, in vitro data, selectivity tests, in vivo/ex vivo experiments, imaging, which make this work really thorough and complete”, and they found the “application and the logic flow of the manuscript to be excellent.”

Below we address in point-by-point fashion the comments and concerns of the Reviewers.

Reviewer #1: The paper by Bolaender et al describes the discovery and validation of new chemical tools that target disease-related (glioblastoma in particular) complexes of proteins. The latter form the so called epichaperomes, complex protein-protein assemblies that form around hsp90 and 70.

The paper is very well written and presents an incredible wealth of data and results. There is a great combination of medicinal chemistry, BBB experiments, in vitro data, selectivity tests, in vivo/ex vivo experiments, imaging, which make this work really thorough and complete. The data also show that the new molecules are selective for GBM and may be useful for mechanistic studies.

The only modifications may be suggested in the discussion, where the authors may want to consider discussing the possibility to use molecules that target PPIs in chaperone complexes that have been developed using different concepts than PU-H71, such as for example the ones in 10.1002/chem.201904523, 10.1002/chem.202000615.

Overall, I appreciated reading this paper, its in-depth investigation from molecular design to in vivo. I think it is worth publication in Nat. Comms.

R1. Response: We thank Reviewer #1 for the recognition of the large effort that went into this work and of its significance of these efforts. We appreciate the suggestion to discuss other modalities to regulate specific protein-protein interaction networks in disease, and have now included such references in the Introduction and References (refs. 4,5).

Reviewer #2: In the manuscript, the authors used chemical tools to overcome the limitation of already known epichaperomes' inhibitor PU-H71. It cannot permeate through blood barrier brain. They found that PU-HZ151 could be a potent chemical probe to pass through BBB and showed the selectivity to epichaperomes. They used isotopes of iodine in the compound structure to do to PET & radiology. For the application, the authors used glioblastoma. Depending on the samples, the formation of epichaperomes were different, but the results showed that PU-HZ151 was selective to the epichaperomes and cell viability. The application and the logic flow of the manuscript are excellent. Therefore, it is a good paper to publish to Nat. Commun. after revising some minor points as below.

[Minor points]

1. I wondered why the authors selected PU-BIS-11 as the final candidate for the replacement of PU-H71, even though it had lower affinity to Hsp90.

R2.1. Response: In addition to target binding affinity, various other properties of candidate probes define the optimum agent for *in vivo* characterization of a molecular target. As illustrated in Figure 1c, the collective properties of our candidate probes led to our current selection.

2. The authors made xenograft mice using MDA-MB-468 and ASPC1. Then they injected only PU-HZ151, but it will be good to compare PU-H71 together to show the better performance of the selected compound. Also, they did not mention that they used muscle and heart from which xenograft mice.

R2.2. Response: We thank you for this comment. Please note, the MDA-MB-468 versus ASPC1 model tests selectivity of the compound for the epichaperome (expressed only in MDA-MB-468) over HSP90 (expressed at

equal levels in both MDA-MB-468 and ASPC1), and not CNS penetration. PU-AD is not expected to be better than PU-H71 in this assay – it is expected to be as good or relatively similar to PU-H71 in this test. Nevertheless, we appreciate the Reviewer's comment and added the PET images for PU-H71 to Supplementary Fig. 5b and include refs.13,15 (Rodina et al. Nature 2016, Pillarsetty et al, Cancer Cell 2019) where we reported on PU-H71's profile.

3. In figure 4C&D, the authors did the autoradiography and H&E staining. The signal of radiography in H&E staining position was weak. Could it be improved?

R2.3. Response: We respectfully disagree, as the autoradiography signal is not “weak.” The Reviewer may be interpreting the mottle/heterogeneity of the signal within the tumor as noise due to a low signal. If that's the Reviewer's interpretation, we would assert that is incorrect and that the heterogeneity of activity reflects actual microscopic anatomic heterogeneity, with hotter areas corresponding to clusters of tumor cells and colder areas to stroma. This anatomic heterogeneity is clearly shown in the enlarged section of the H&E image. Note further that the autoradiogram is near-optimally exposed - with the hottest area corresponding to the top of the color-scale bar. Attempting to “intensify” the signal with a longer exposure would saturate the image and make it non-quantitative. We added the following sentence to the Fig. 4 figure legend for clarification “The relative area occupied by the brain tumor is shown with a blue line. Inset in (d) shows the heterogeneity of the tumor with clusters of cancer cells surrounded by stroma.” We have also added explanatory notes on the Figure itself.

4. In figure 5C&D, the information is missing which PU compound was used.

R2.4. Response: The information is now added as suggested.

Reviewer #3: The HSP90 inhibitor, or now “epichaperome complex”, has been proposed for use in cancer and AD for some time. This work offers new methods to screen for compounds that hit targets within the “epichaperome”. This is an interesting framework for imaging a compound that selects for tumors based on structural differences in the arrangement of common chaperone proteins that cause the agent to reside longer in the malignant regions. Claims for applications in the CNS and AD are less convincing than the work towards exploring the hypotheses in oncology. There are a few concerns that need to be addressed prior to publication:

R3.1. Response: We appreciate the Reviewer's interest in our work but respectfully disagree with the statement “HSP90 inhibitor, or now “epichaperome complex”, has been proposed for use in cancer and AD for some time”. An important facet of our work is a fundamental - structural, dynamic, and functional - difference between HSP90 and the epichaperome as we have demonstrated in a series of publications on the subject (Rodina et al., Nature 2016; Kishinevsky et al., Nature Communications 2018; Pillarsetty et al., Cancer Cell 2019; Inda et al., Nature Communications 2020; and Yan et al., Cell Reports 2020, refs. 13-17). Please note that epichaperomes are scaffolds – they rewire the connectivity and function of protein networks by remodeling how thousands of proteins interact in conditions of chronic cellular stress (for example in diseases such as cancers, as well as Alzheimer's disease and Parkinson's disease). Conversely, HSP90 is a chaperone. Chaperones, co-chaperones, and their complexes have defined functions as folders – they interact with a protein to process it through the chaperone folding cycle. Epichaperomes are long-lived oligomers of chaperome members. This differs from chaperones such as HSP90, which interact in a highly dynamic manner with one another and with client proteins on the millisecond to second timescale to make folding versus degradation decisions through transient protein-protein interactions within the context of the proteostasis network that are central to maintaining the cellular proteome. Epichaperomes are specific to cells exposed to defined stressors. Conversely, chaperones are highly abundant and ubiquitous proteins. Therefore, chaperones and epichaperomes are distinct and targeting the epichaperome should not be confused with targeting HSP90 *per se*.

Major concerns:

1. The authors claim that they have developed probes to modulate the epichaperome is too broad and needs to be scaled back from the title, abstract and throughout the manuscript, as the molecules are still designed to bind to one specific target (HSP90).

R3.2. Response: Please note the target of our compounds and probes is the epichaperome associated HSP90 (found only in specific cells), not the HSP90 pools involved in folding and other housekeeping functions (found abundantly in all cells in the mouse or human body). Our previous publications have established the structural and functional difference between the epichaperome and HSP90 (Rodina et al., Nature 2016; Kishinevsky et al., Nature Communications 2018; Pillarsetty et al., Cancer Cell 2019; Inda et al., Nature Communications 2020; Yan et al., Cell Reports 2020; Sugita et al., Nature Precision Oncology 2020; and Jhaveri et al., JCO Precision Oncology 2020, refs. 13-17, 48 and 54), and we believe the title, abstract, and text correctly represent these scientific discoveries and developments, particularly in the context of what we illustrate in the current report.

The testing platform we introduce here springboards from these discoveries to demonstrate, both biochemically and functionally, epichaperome over HSP90 selectivity, as we detail below:

Biochemical validation of the selectivity of the ligands for epichaperome over HSP90:

Figure 3a-c: Please note, both the MDA-MB-468 and ASPC1 tumors express equal levels of HSP90, yet only MDA-MB-468 shows epichaperome positivity. PU-HZ151 selects for the epichaperome over HSP90 as evidenced by selective retention in epichaperome-positive tumors over epichaperome-negative tumors (see Fig. 3b,d) despite equal delivery of PU-HZ151 in both tumors (see time 0 to 30 min post-injection).

Figure 4: Both the cancerous lesions and the non-affected brain regions express high HSP90 levels, yet only the intracranial MDA-MB-468 is epichaperome positive. PU-HZ151 selects for the epichaperome over HSP90.

Figure 5b: All the patient derived glioma stem cells express high, equal levels of HSP90 (Fig. 6g, Supplementary Fig. 6); yet only the epichaperome-positive tumors show PU-HZ151 binding (Fig. 5b and Fig. 7c,d)

Functional validation of the selectivity of the ligands for epichaperome over HSP90:

Figure 6e,g and 7: All the patient derived glioma stem cells express high, equal levels of HSP90 (Fig. 6g, Supplementary Fig. 6), yet only the epichaperome-positive tumors show apoptotic sensitivity to PU-HZ151. There is a positive correlation between apoptotic sensitivity to PU-HZ151 and epichaperome expression, a hallmark of drug action via epichaperomes, as we reported previously (Rodina et al., Nature 2016; Kourtis et al., Nature Medicine 2018). There is no positive relationship between sensitivity of these primary specimens to PU-HZ151 and the expression of HSP90 client proteins (Supplementary Fig. 6), a hallmark of drug action via HSP90 as reported (PMID: 28679777).

Lastly, if the *in vivo* target of our probe would be the chaperone HSP90, we would get high non-specific uptake not only in all epichaperome-positive tumors but also in epichaperome-negative tumors and in all tissues and organs because HSP90 is one of the most abundantly expressed proteins in a cell (~2-4% of the total protein mass in both dysfunctional and normal cells, and amounting to 200-300g in an adult that weighs 70kg, estimating for a total of 16% protein mass), which is not what we observe. Rather uptake is highly specific.

Specifically, the lead compound “151”, appears to have some use in oncology but the applications in AD are not clear. Although the probe is primarily selected for brain penetration, most of the work was carried out in tumors (including the GBM model which was peripheral). It is not justified based on the current experiments to claim that this is a “CNS-directed epichaperome probe”.

R3.3. Response: The GBM models are not peripheral – these are **orthotopic** tumors, i.e., established by intracranial injection of glioma stem cells, and we highlight throughout the text and in the methods that such is the case. These tumors mimic the human disease, such as brain invasion, angiogenesis, and resistance to therapy, as reported [PMID: 29392700].

The claim for “CNS-directed epichaperome probe” is supported by a wealth of experimental evidence:

- a. **Figure 2, excellent delivery into non-diseased brain at par with plasma and non CNS- and non-Pgp organ and tissue exposure** (these figures were initially supplementary figures, now have been incorporated into Figure 2a,b).

- b. **Figure 4**, delivery into, retention in and selectivity for an epichaperome-positive model of breast cancer metastasis to the brain (thus in brain).
- c. **Figure 5**, delivery into, retention in and selectivity for epichaperome-positive orthotopic gliomas (in brain) derived from glioma stem cells and **Figure 7**, functional engagement of the target, the epichaperome, in epichaperome-positive orthotopic gliomas (demonstrating target engagement in brain).

With regards to Alzheimer's disease (AD), this paper's aim is not to recapitulate previous findings. Rather, our stated goal is to "provide structural, biochemical, and functional insights into the discovery of epichaperome probes, with a focus on epichaperome probes for use in central nervous system diseases". We already published the discovery of epichaperomes as targets for AD (Inda et al., Nature Communications 2020, ref.16 and highlighted by an NIH press release <https://www.nia.nih.gov/news/faulty-protein-connections-short-circuit-brain-alzheimers-disease>).

a. There is a lack of evidence that this approach will be clinical useful in the central nervous system as claimed in the title and throughout the manuscript. For example, what is the ligand binding to in AD? Authors need to also address toxicity issues and lack of in vivo efficacy from this approach.

R3.4. Response: The target is the epichaperome as stated. We show using a battery of tests, and as detailed throughout the manuscript, epichaperome selectivity and activity through the epichaperome target.

We respectfully disagree with the comment regarding "lack of evidence that this approach will be of clinical use". In addition to activity we show here in target-positive, aggressive orthotopic patient-derived gliomas, PU-HZ151 has successfully completed IND-enabling studies, has received FDA approval to move into Phase 1 clinical studies in healthy volunteers, has successfully completed such Phase 1 and is now in Phase 2 in Alzheimer's disease under the clinical name of PU-AD. We believe this to be good evidence for "clinical usefulness".

It is unclear what "lack of efficacy" the Reviewer is referring to. If this is in reference to the GBM models that do not express the target, that is not "lack of efficacy". Rather, it is proof of on-target activity and selectivity, as the agent shows activity only in the GBMs that express the target and no activity in GBMs that do not express the epichaperome target (but do express abundant HSP90 and HSP90 client proteins). Conversely, the Reviewer may be referring to a "lack of efficacy" from epichaperome targeting, which is a perplexing comment considering that epichaperome targeting in cancer by PU-H71 has shown both safety and efficacy in historically difficult to treat cancers such as triple-negative breast cancer and myeloproliferative neoplasm transformed to acute myeloid leukemia (Sugita et al., Nature Precision Oncology 2020 and Jhaveri et al., JCO Precision Oncology 2020, refs. 48 and 54).

b. Standard neuroimaging evaluations for a new radiopharmaceutical were not considered and need to be included to make claims about CNS and AD. For example:

i. The uptake in tumors and low uptake in brain is demonstrated. However, specificity is questionable in the CNS in vivo. Why were blocking experiments not performed, in the in vitro autoradiography or in vivo imaging? These are required and should be included.

R3.5. Response: Our extensive data clearly demonstrate the molecular specificity of the probe, as we also detailed in our response to query **R3.2**. Performing binding in target-positive versus target-negative models is a more powerful approach to study selectivity than is to do blocking studies (see detailed response on this point below).

The comment of "low uptake" in the brain is perplexing. There are three key parameters that combined determine "brain uptake" AND "target engagement in the CNS":

1. The PK and the exposure of the brain lesions to the chemical probe (to understand brain delivery).
2. The ratio of probe's presence in target-positive lesions over normal brain and target-negative lesions (to understand selectivity and specificity, i.e. no binding to target negative brain structures).
3. The functional engagement of the target positive lesions over the target negative lesions (to demonstrate productive target engagement).

Though “uptake” determines technically if a compound enters brain tissue, this parameter alone is inadequate to determine if such brain penetration results in ligand–target interaction. Our strategy covers all three aspects, as we detail below, to show *i. excellent PK, ii. specific exposure of the target, and iii. functional engagement of the target in the target-positive lesions over the normal brain and target-negative lesions:*

- 1. The PK and the exposure of the brain lesions to the chemical probe (to understand brain delivery).**
 - i.* The brain delivery of PU-HZ151 in both mice and humans is **at par with its delivery into non CNS- and non-Pgp organs** (see C_{max} delivered to brain and lung in Figure 2a is **10-15 μM** upon single intravenous injection of 10mg/kg PU-HZ151). See Figure 2b, the AUC for brain compares to that of lung and muscle. *ii.* The brain to plasma ratio is **8.25/3.52 = 2.35**. *iii.* The concentration retained in the epichaperome-positive brain lesions at **24 h** post single intravenous injection of 50mg/kg (the dose used in the GBM PDX studies from Figure 7) is **1 μM (i.e. epichaperome saturating)**. The uptake of PU-HZ151 in the brain is therefore excellent not low.
- 2. The ratio of probe’s presence in target-positive lesions over normal brain and target-negative lesions (to understand selectivity and specificity, namely no binding to target negative brain structures).** With regard to specificity, we demonstrate when mixed with the thousands of proteins as present in the mouse body, the probe engages one target – the epichaperome. Abundant evidence is presented in Figs. 1-7, biochemically and functionally. We agree, blocking studies are conducted frequently but these only give a hint at specificity. If administration of a cold therapeutic dose prior to tracer injection can decrease tracer binding, this only means that the cold competition blocker occupies binding sites that the tracer cannot bind to. Therefore, we prefer instead to use *in vitro* and *in vivo* models of **both target-positive and target-negative lesions** to demonstrate specificity, which we have done beyond reasonable doubt throughout the revised manuscript. See also newly added blocking study to Figure 1e.
- 3. The functional engagement of the target positive lesions over the target negative lesions (to demonstrate productive target engagement).** Please refer to Figs. 6, 7 and their associated Supplementary Figures, which are designed to address both *in vitro* and *in vivo* such on-target activity and functional target engagement.

ii. The authors need to show selectivity over other CNS targets *in vitro* (not just potency to the primary target) for neuroimaging in AD.

R3.6. Response: It is unclear how screening for target selectivity *in vitro* – screening against other CNS proteins confirms or refutes the specific binding of our probes, as demonstrated here abundantly *in vivo* at the whole-body level (see also **R3.2**). Nonetheless, we added the Caliper GenSep II screen and the DiscoverX scanEDGE Profiling Service, both ran at **10 μM** PU-HZ151, as requested by the Reviewer. These are attached as Supplementary files (see Supplementary Fig.2 and Supplementary Table 7). The following text was added to the manuscript: ‘PU-HZ151 was assayed *in vitro* at 10 μM for off-target binding to 167 substrates (70 in the General Side Effect Profile (SEP) II diversity panel and 97 in the scanEDGE kinase panel) to show little to no off-target binding interactions (Supplementary Information). General SEP II contains brain-resident key proteins, including neurotransmitter receptors, ion channels, ion pumps, synthetic enzymes, and transporter proteins. ScanEDGE focuses on kinases and was used to exclude potential non-specific interaction of PU-HZ151 (a purine-scaffold agent) with typical ATP-binders, such as kinases.’

iii. Brain penetration is low (ca. 1 SUV) and could have been improved. If BBB penetration is a major focus of the medicinal chemistry, the authors would benefit from CNS MPO analyses and should be reported. The authors focused on solubility and lipophilicity (but only logP or logD are discussed and should be discussed in the context of neuro-PET tracers that have log D in the 1.5-3.5 range. The authors have minimal data reported for solubility; “acid/base character” is normally called apparent pKa, which is used later in the same paragraph; there is no mention of efflux transporter such as Pgp in this section either and such assays should also be reported to give insights into the medicinal chemistry design for crossing the BBB.

R3.7. Response: Please see above our response to “brain penetration is low” critique in response **R3.5**.

A paragraph regarding log D was added to the text: “PU-HZ151 has a logD of 2.37 as compared to 1.21 for PU-H71, Supplementary Table 6”. With regards to the CNS MPO parameters and others, these are included in the Supplementary files, now as Supplementary Table 6. CNS MPO parameters aim for extended plasma

circulation/stability in liver microsomes/low metabolic clearance as a mechanism for obtaining good brain exposure. As we demonstrate here and in our prior publications, in the discovery and development of epichaperome agents we do not aim for extended presence of the agent at sites other than those expressing the target (Rodina et al., Nature 2016; Kishinevsky et al., Nature Communications 2018; Pillarsetty et al., Cancer Cell 2019; Inda et al., Nature Communications 2020; and Yan et al., Cell Reports 2020, ref.13-17 and detailed in the critique response **R3.2**). The slow off-rate of the agent from the target assures that, once the ligand binds to the target, the target remains blocked for days. Therefore, a good C_{max} at the site of target expression and a rapid systemic clearance, are the two parameters we aim for, as we recently reported (Pillarsetty et al., Cancer Cell 2019, ref.15). A summary of the above paragraph was added to the text.

We added P-glycoprotein studies to Supplementary Fig. 3 to show PU-HZ151 is not a Pgp substrate. We added to the text: 'PU-HZ151 showed comparable activity in both P-glycoprotein (Pgp)-overexpressing and Pgp-low epichaperome-positive cell lines, suggesting its efficacy is unlikely to be affected by Pgp overexpression in the brain (Supplementary Fig. 3a-c).'

iv. Radiometabolite analysis needs to be done and reported.

R3.8. Response: Metabolite analysis of PU-HZ151 was added to the Supplementary Information, Supplementary Table 8,9 and Supplementary Fig. 4. Please note however, the only species present at and retained in the brain lesions (where the target is) is the intact molecule. The value of metabolite analysis therefore appears minimal in the context of using these agents as chemical probes for brain-resident diseases. The following text was added to the manuscript: 'The metabolic stability and metabolite characterization of PU-HZ151 were evaluated, and the cytochrome P450 enzymes responsible for the metabolism of PU-HZ151 were identified (Supplementary Table 8,9 and Supplementary Fig. 4a-c). Six putative metabolites, two major and four minor, were identified in liver microsomes of human, rat and dog and three metabolites in mouse microsomes. PU-HZ151 is mainly metabolized by 3A, 3A4 and 2C9 and to a lesser extent by 1A2, 2D6, 2B6, 2C8, and 2C19 and is essentially unaffected by 2A6. However, we detected only the intact PU-HZ151 molecule in epichaperome-positive lesions and in the brain after 1 h post-intravenous PU-HZ151 injection, indicating metabolites are both rapidly cleared from the brain and the whole body and/or are epichaperome-inert (Supplementary Fig. 4a-c).'

c. In Figure 3b, the Native-PAGE gel (right panel) shows that bands appear for both MB-468 and ASPC-1 samples, with MB-468 cells having a much higher signal compared to ASPC-1, however, without a loading control or any indication of how much total protein was loaded the reader cannot be certain that the MB-468 sample isn't overloaded. The authors state that MB-468 has high expression of epichaperomes while ASPC-1 has low expression, however, there is no loading control probe, such as β -actin to demonstrate that the same amount of sample was loaded into each sample well, as seen in the SDS/PAGE gel. The authors should indicate the amount of total protein loaded into each sample lane to clarify whether the same total protein quantities were used, and should have a loading control to indicate these quantities. The same concerns is noted in the Native-PAGE blot in Figure 6c.

R3.9 Response: The SDS PAGE located next to the NATIVE PAGE demonstrates the loading. Equal amounts of material were loaded for each cell line and to both Native and SDS PAGE. We demonstrated through a multitude of alternative methods that MDA-MB-468 is an epichaperome-positive cancer cell whereas ASPC1 is epichaperome-negative (Rodina et al., Nature 2016, ref.13). Nevertheless, we now also add β -actin blotting to the NATIVE PAGE in Fig. 3b.

Minor concerns:

1. The chemistry was done on a small scale which is acceptable for screening purposes, however, the characterization is incomplete for the standard of Nature or Medicinal Chemistry journals. Specifically, several compounds are missing ^{13}C NMR spectra, HRMS, etc.

R3.10. Response: Thank you for catching these missing pieces of information. We have carefully checked the methods section and added the missing ^{13}C NMR spectral data in the revised submission. As per the instructions, HRMS data are preferred but not required and we provide either HRMS or MS data, as available.

2. The nomenclature consensus guidelines should be used for this manuscript (e.g. consistent use of square brackets when needed, mCi vs. Bq, etc).

R3.11. Response: We revised the manuscript to ensure consistent nomenclature throughout. Square brackets were added where appropriate and Bq was transformed into mCi in the one instance it was used in the Methods section.

3. The authors describe the compounds as "probes" even when unlabeled. Are the cold compounds to be used as a tool/probe just to study the epichaperome or therapeutic (if so the authors should justify why a compound that binds to the epichaperome proteins would be expected to be therapeutic). The use of this probe/therapeutic agent may be limited to understanding the role of the epichaperome in tumour vitality. A probe directed at tumour stem cells would still be necessary to determine the efficacy of the treatment.

R3.12. Response: Both the cold compound and its labeled version are chemical probes, as discussed throughout the manuscript. Figure 8 was added in clarification. The cold compound is also a potential therapeutic candidate - as mentioned in the text, this agent under the clinical name PU-AD, is currently in Phase 2 clinical evaluation in human AD. Why the epichaperome is a target in both cancers and AD is referenced and discussed in the **Introduction** and **Discussion**. As requested, we expanded this topic in the **Discussion**. The comment regarding the 'A probe directed at tumour stem cells would be needed' is unclear. The probe binds to a target not to cells, and if the target is expressed in cells of the tumor microenvironment or others, the probe will recognize them.

4. 3) It is not clear whether the dose required for therapy would be detrimental to the "healthy" form of the chaperone proteins. The concentration sufficient to cause apoptosis in the explants is only specified in Fig 5 caption but not the Methods.

R3.13. Response: The therapeutic index in our approach to epichaperome therapy is gained through an important principle: the agents have an off-rate from "the healthy" chaperone that is very fast whereas the off-rate from the epichaperome is very slow (i.e. minutes versus days, as demonstrated extensively in this manuscript for both the 'cold PU-HZ151 via LC/MS-MS PK measurements and for the 'tracer' [¹²⁴I]-PU-HZ151 via imaging. Therefore, in the *in vivo* setting the 'healthy' chaperone is not functionally engaged during treatment (see **Discussion** and relevant references throughout the text). This was also reported and abundantly demonstrated also for the non-CNS agent PU-H71 (Rodina et al., Nature 2016; Pillarsetty et al., Cancer Cell 2019; Sugita et al., Nature Precision Oncology 2020; and Jhaveri et al., JCO Precision Oncology 2020, refs. 13,15, 48 and 54). We also updated Fig. 1 to show the concentration-dependent effect of PU-HZ151 (see Fig. 1e-g).

5. The abstract is very vague and could benefit from being rewritten. The title should be revised to eliminate the implication that these compounds are designed for the CNS and should also be more specific to the experiments conducted herein.

R3.14. Response: We respectfully disagree as we have provided abundant evidence on the CNS utility, a sentiment also expressed by Reviewers 1 and 2.

6. The preclinical MRI scanner make and model, and the weighting of the MRI images (presume T2-weighted) should be added to the experimental details.

The sentence "Magnetic resonance imaging (MRI), by offering structural and soft tissue resolution, confirmed the presence of PU-HZ151 in the brain tissue" is misleading. MRI is only providing anatomical detail to co-localize with the PET signal, but this sentence, depending on which word is stressed, implies that MRI is sensitive to the compound.

R3.15. Response: We provided the requested information in the revised Methods, as suggested. We also edited the text to eliminate ambiguity, as suggested.

REVIEWERS' COMMENTS

Reviewer #1 (Remarks to the Author):

The authors have done a good job at further improving an already great paper, which obtained good recognition also by the other 2 reviewers. I recommend it for publication.

References 4 and 5 report a wrong journal name: it should be just "Chemistry" or "Chemistry - A European Journal" and NOT "Chemistry (Easton)"

Reviewer #2 (Remarks to the Author):

All the minor concerns I raised to the first version of the paper have been properly addressed. So, I feel the paper is ready to be accepted.

Reviewer #4 (Remarks to the Author):

The radiolabel and injection of the agent into humans does not clearly demonstrate the uptake. If the half-life of the agent is 4 days and the protein binds to the DNA, then there should be signal at 24 hours, but in both cases there is none. It is possible that the control agent is cleared as first pass in the liver, and there may be a differential in the AUC that can be measured, but it does not seem to show cellular uptake in the brain cells. This short window for the lung tumor also seems like it would be visible beyond 0.5h if it was binding to the tumor cell DNA rather than being a vascular phenomenon.

The blood brain barrier in animal tumors is not 'intact' although there are some models where there is variability in the BBB based on MR and that can be valuable. The assessment of dissociated, processed tumor cells in vitro is probably not the best test to detect a subtle difference in efficacy - might need a control using TMZ.

We thank the Reviewers for their thoughtful and constructive critiques during the review process. We are pleased Reviewers #1 and #2 concluded this manuscript is ready for publication in *Nature Communications*.

Below we address the remaining concerns of Reviewer#4.

Reviewer #1:

The authors have done a good job at further improving an already great paper, which obtained good recognition also by the other 2 reviewers. I recommend it for publication.

References 4 and 5 report a wrong journal name: it should be just "Chemistry" or "Chemistry - A European Journal" and NOT "Chemistry (Easton)"

R1. Response: We thank Reviewer #1 for the very kind words. We apologize we missed the EndNote error for refs. 4,5 and will make sure it is corrected.

Reviewer #2:

All the minor concerns I raised to the first version of the paper have been properly addressed. So, I feel the paper is ready to be accepted.

R2. Response: Thank you for appreciating our diligence in responding to critiques and for deeming this paper is ready to be accepted.

Reviewer #4:

The radiolabel and injection of the agent into humans does not clearly demonstrate the uptake. If the half-life of the agent is 4 days and the protein binds to the DNA, then there should be signal at 24 hours, but in both cases there is none. It is possible that the control agent is cleared as first pass in the liver, and there may be a differential in the AUC that can be measured, but it does not seem to show cellular uptake in the brain cells. This short window for the lung tumor also seems like it would be visible beyond 0.5h if it was binding to the tumor cell DNA rather than being a vascular phenomenon.

The blood brain barrier in animal tumors is not 'intact' although there are some models where there is variability in the BBB based on MR and that can be valuable. The assessment of dissociated, processed tumor cells in vitro is probably not the best test to detect a subtle difference in efficacy - might need a control using TMZ.

R4. Response: Thank you for your comments. Is it possible the Reviewer is referring to the *physical* half-life of ^{124}I , which is 4 days, and not to the *effective* half-life which is determined by target binding. The effective half-life of the isotope (as part of ^{124}I -PU-HZ151 probe) in brain tissue is determined also by the biological half-life - ie, how rapidly ^{124}I -PU-HZ151 is cleared from the tissue-of-interest that does or does not contain the target. In the case of normal health brain tissues, which lack the epichaperome-target, ^{124}I -PU-HZ151 has nothing to which to bind and hence clears rapidly; ie, the biological half-life is very brief, less than 0.5 hours; and, hence, the effective half-life is also very short.

This is also the case for normal lung tissue – no target – no retention. Conversely, for the lung tumor (ie target expressing), the PET image and the quantification of the PET signal show clear ^{124}I -PU-HZ151 probe retention at 3 h and 24 h (thus beyond the 0.5 h recorded for normal, non-target expressing, lung tissue, Fig. 3g,h). Thus there is clear difference in the half-life of the probe in target-expressing versus target non-expressing tissues, which our experiments throughout the manuscript confirm is on- and through-target.

Regarding “vascular phenomenon”: If such were the case, we should see indiscriminate uptake and retention of the probe in all the in vivo models under investigation, which we do not.

Regarding “the BBB of tumor bearing mice is not intact”: Thank you for the comment. Such is certainly true. That is exactly why we use both target-positive and target-negative orthotopic gliomas to confirm the on-target specificity and activity of the probe.

In sum, the brain uptake AND the retention of the probe in target-expressing brain lesions over the non-target expressing brain lesions and non-target expressing normal brain is supported here by a multitude of biochemical, analytical and functional methods, both in cells and mice, with proof-of-principle evidence in humans. We added a sentence to the Discussion section to capture the gist of our response to Reviewer#4 regarding the radio-labelled tracer uptake.